# Basolateral amygdala oscillations enable fear learning in a biophysical model

**Anna Cattani[1]\*, Don B Arnold[2], Michelle McCarthy[1], Nancy Kopell[1]**

[1]Department of Mathematics and Statistics, Boston University, Boston, United States;
[2]Department of Biology, University of Southern California, Los Angeles, United States

## eLife Assessment

This **valuable** modeling study explores how biophysical properties of different interneuron subtypes in the basolateral amygdala (BLA) enable production of oscillations that facilitate functions such as spike-timing-dependent plasticity. Simulated networks provide **solid** evidence that highlights the importance of interactions between interneurons for some forms of spike-timing dependent plasticity. This work will likely be of interest to investigators studying interactions among interneurons, rhythms in the amygdala, and mechanisms of plasticity thought to underlie associative learning.

**Abstract** The basolateral amygdala (BLA) is a key site where fear learning takes place through synaptic plasticity. Rodent research shows prominent low theta (~3–6 Hz), high theta (~6–12 Hz), and gamma (>30 Hz) rhythms in the BLA local field potential recordings. However, it is not understood what role these rhythms play in supporting the plasticity. Here, we create a biophysically detailed model of the BLA circuit to show that several classes of interneurons (PV, SOM, and VIP) in the BLA can be critically involved in producing the rhythms; these rhythms promote the formation of a dedicated fear circuit shaped through spike-timing-dependent plasticity. Each class of interneurons is necessary for the plasticity. We find that the low theta rhythm is a biomarker of successful fear conditioning. The model makes use of interneurons commonly found in the cortex and, hence, may apply to a wide variety of associative learning situations.

**\*For correspondence:**
acattani@bu.edu

**Competing interest:** The authors declare that no competing interests exist.

## Introduction

Pavlovian fear conditioning is widely used as a model of associative learning across multiple species (*Phelps and LeDoux, 2005*) and has been used more generally to study plasticity and memory formation. The major open questions relate to how plasticity is instantiated in defined circuits and how it is regulated by the circuit components (*Rumpel et al., 2005*; *Sah et al., 2008*; *Johansen et al., 2014*; *Bocchio et al., 2017*; *Grewe et al., 2017*). The fear conditioning paradigm consists of a neutral stimulus (conditioned stimulus, CS) presented one or more times together with an aversive stimulus (unconditioned stimulus, US), which induces a fear response. The presentation of the US and CS may or may not overlap in time (e.g. see *Laxmi et al., 2003*; *Stujenske et al., 2014*; *Krabbe et al., 2019*).

In the basolateral amygdala (BLA), the main site of fear learning in the mammalian brain (*Fanselow and LeDoux, 1999*; *Tovote et al., 2015*; *Krabbe et al., 2018*), local field potential recordings (LFP) show prominent low theta (~3–6 Hz), high theta (~6–12 Hz), and gamma (>30 Hz) rhythms (*Seidenbecher et al., 2003*; *Courtin et al., 2014b*; *Stujenske et al., 2014*; *Davis et al., 2017*). Recent rodent experimental studies (*Antonoudiou et al., 2022*; *Bratsch-Prince et al., 2024*) suggest that BLA can intrinsically generate theta oscillations (3–12 Hz). Furthermore, other studies show increased low theta (*Davis et al., 2017*) and gamma (*Courtin et al., 2014b*) in LFP recordings after successful fear conditioning, whereas modulation of high theta is associated with fear extinction (*Davis et al., 2017*), a

paradigm that aims to suppress the association between CS and fear (*Bouton, 2004*); the modulation of the power of these rhythms suggests they may be associated with BLA plasticity. However, the potential roles of rhythms in instantiating plasticity needed for successful learning are still under investigation (*Bocchio et al., 2017*). We also note that there is not uniformity on the exact frequencies associated with low and high theta, for example (*Lorétan et al., 2004*) used 2–6 Hz for low theta. Here, we use 2–6 Hz for the theta range and 6–14 Hz for the high theta range.

In this paper, we aim to show (1) How a variety of BLA interneurons (parvalbumin-expressing [PV], somatostatin-expressing [SOM], and vasoactive intestinal peptide-expressing [VIP]) lead to the creation of these rhythms and (2) How the interaction of the interneurons and the rhythms leads to the appropriate timing of the cells responding to the US and those responding to the CS to promote fear association through spike-timing-dependent plasticity (STDP). Since STDP requires overlap of the effects of the CS and US, and some conditioning paradigms do not have overlapping US and CS, we include as a hypothesis that the effects of the CS and US overlap even if the CS and US stimuli do not. In the Discussion, we suggest how neuromodulation by ACh and/or dopamine can provide such overlap. We create a biophysically detailed model of the BLA circuit involving all three types of interneurons and show how each may participate in producing the experimentally observed rhythms and interacting to produce the necessary timing for fear learning. In particular, we find that the low theta, high theta, and gamma rhythms in the BLA originating from the BLA interneurons promote the formation of a dedicated fear circuit shaped through rhythmic regulation of depression-dominated STDP. In this model, if any of the classes of interneurons are removed from the circuit, the rhythms are changed and there is a failure of the plasticity needed for successful learning. We show that fine timing between excitatory projection neurons responding to the CS and US is necessary but not sufficient to produce associative plasticity when the latter can be affected by the entire history of spikes, not just the most recent pre- and post-synaptic spikes. The other critical element for plasticity is the interaction of interneurons that creates pauses in excitatory cell activity.

The model reproduces the increase in the low theta after training that was found in the experimental data (*Davis et al., 2017*). This increase was not seen for network instantiations that did not learn; thus, the simulations suggest that the increase in low theta is a biomarker of successful fear conditioning. Furthermore, this low theta signal is a consequence of network dynamics that emerge due to the newly formed synapse linking CS-encoding and fear-encoding neurons. Finally, we note that the ideas in the model may apply very generally to associative learning in the cortex, which contains similar subcircuits of pyramidal cells and interneurons: PV, SOM, and VIP cells.

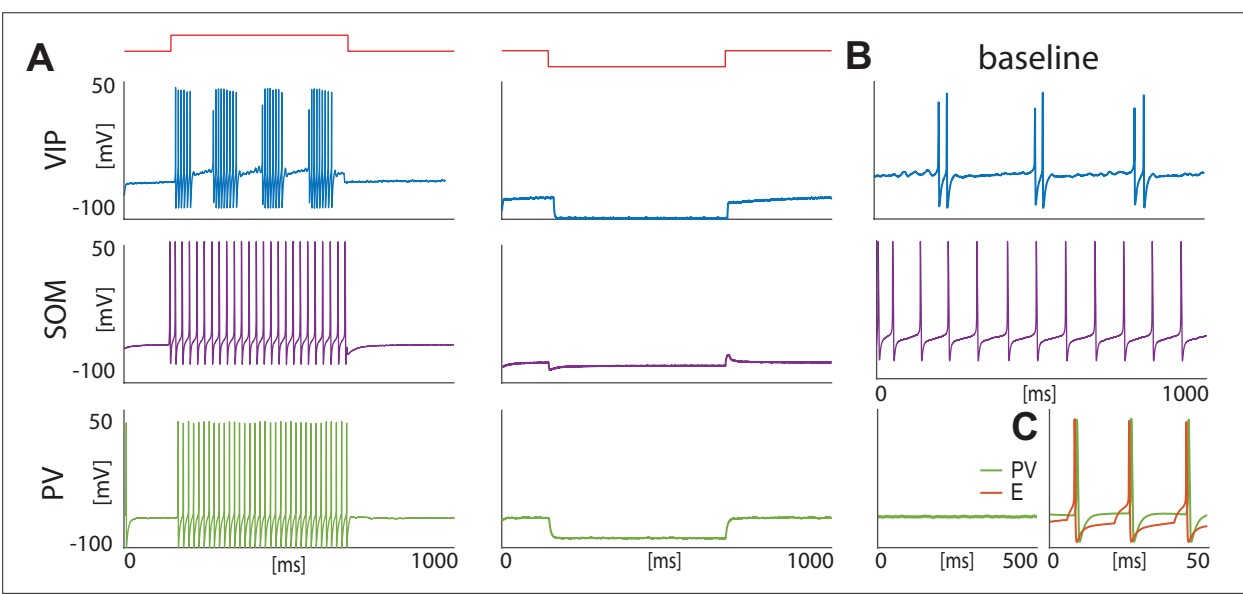

**Figure 1.** Isolated neurons produce fundamental rhythms. (**A**) dynamics in response to depolarizing and hyperpolarizing currents mimic the electrophysiological behavior of BLA interneurons classified in *Sosulina et al., 2010*. (**B**) dynamics in the baseline condition. (**C**). Interacting PV and excitatory projection neuron (**E**) entrain in a pyramidal-interneuron network gamma rhythm (PING).

## Results

### Rhythms in the BLA can be produced by interneurons

Brain rhythms are thought to be generated and propagated largely by interneurons (**Whittington et al., 2000**). Identified interneurons in BLA include VIP, SOM, and PV (**Muller et al., 2006**; **Muller et al., 2007**; **Rainnie et al., 2006**; **Bienvenu et al., 2012**; **Krabbe et al., 2018**; **Krabbe et al., 2019**), which can be further subdivided according to their electrophysiological dynamics (**Sosulina et al., 2010**; **Spampanato et al., 2011**). In the model, we show that some types of VIP, SOM, and PV can each contribute to the generation of a key rhythm involved in the BLA due to their specific intrinsic currents.

For VIP interneurons, we consider a subtype that responds to a depolarizing step current with bursting or stuttering behavior (**Sosulina et al., 2010**; **Spampanato et al., 2011**). This type of behavior can be elicited by a D-type potassium current, a current thought to be found in a similar electrophysiological subtype of VIP interneurons in the cortex (**Porter et al., 1998**). In our model, VIP interneurons endowed with a D-current respond to depolarizing currents with long bursts, and to hyperpolarizing currents with no action potentials, thus reproducing the electrophysiological properties of type I BLA VIP interneurons in **Sosulina et al., 2010** (**Figure 1A**, top): in the baseline condition, the condition without any external input from the fear conditioning paradigm (**Figure 1B**, top), our VIP neurons exhibit short bursts of gamma activity (~38 Hz) at low theta frequencies (~2–6 Hz; peaking at ~3.5 Hz; see **Appendix 1—figure 1A**).

For SOM interneurons, we focus on the electrophysiologic behavior of type III BLA SOM cells in **Sosulina et al., 2010**, showing regular spikes with early spike-frequency adaptation in response to a depolarizing current, and pronounced inward rectification (downward deflection) and outward rectification (upward deflection) upon the initiation and release of a hyperpolarizing current. SOM interneurons have been mostly studied in the hippocampus, especially the O-LM cells (**Maccaferri and McBain, 1996**; **Gillies et al., 2002**; **Saraga et al., 2003**; **Rotstein et al., 2005**), which are known to have a hyperpolarization-activated current, that is H-current, and a persistent sodium current,that is NaP-current. An H-current has also been observed in the SOM cells in the BLA (**Ünal et al., 2020**). In our model, with the introduction of both H- and NaP-currents with specific conductances (see Materials and methods for details), the SOM cells mimic the electrophysiologic behavior of type III BLA SOM cells in **Sosulina et al., 2010** (**Figure 1A**, middle). In our baseline model, SOM cells have a natural frequency of ~12 Hz (**Figure 1B**, middle; **Appendix 1—figure 1B**), which is at the upper limit of the experimental high theta range; this motivates our choice to extend the high theta range up to 14 Hz in order to include the peak.

Our model PV interneurons are fast-spiking interneurons (FSIs) with standard action potentials produced by Hodgkin-Huxley-type sodium and potassium conductances. They are silent at baseline condition (**Figure 1B**, bottom) and show similar behaviors to type IV interneurons (**Sosulina et al., 2010**) in response to depolarizing and hyperpolarizing currents (**Figure 1A**, bottom). However, when reciprocal connections are present between the PV interneuron and the excitatory projection neuron, these neurons form a PING rhythm (pyramidal-interneuron network gamma; **Whittington et al., 2000**), if the excitatory projection neuron receives enough excitation to fire (**Figure 1C**); this has been suggested as a possible mechanism for the basis of gamma rhythm generation in the BLA (**Feng et al., 2019**). The frequency of PING depends sensitively on the external input to the excitatory projection neuron and the PV's decay time constant of inhibition (8.3 ms).

### Interneurons interact to modulate fear neuron output

Our BLA network consists of interneurons, detailed in the previous section, and excitatory projection neurons (**Figure 2A**). Both the fear-encoding neuron (F), an excitatory projection neuron, and the VIP interneuron are activated by the noxious stimulus US (**Krabbe et al., 2019**). The US input on the pyramidal cell and VIP interneuron is modeled as a Poisson spike train at ~50 Hz and an applied current, respectively. In the rest of the paper, we will use the words 'US' as shorthand for 'the effects of US'. As shown in **Figure 2A** (top, right), VIP disinhibits F by inhibiting both SOM and PV, as suggested in **Krabbe et al., 2019**. We do not include connections from PV to SOM and VIP, nor connections from SOM to PV and VIP, since those connections have been shown to be significantly weaker than the ones included (**Krabbe et al., 2019**). The simplest network we consider is made of one neuron for each

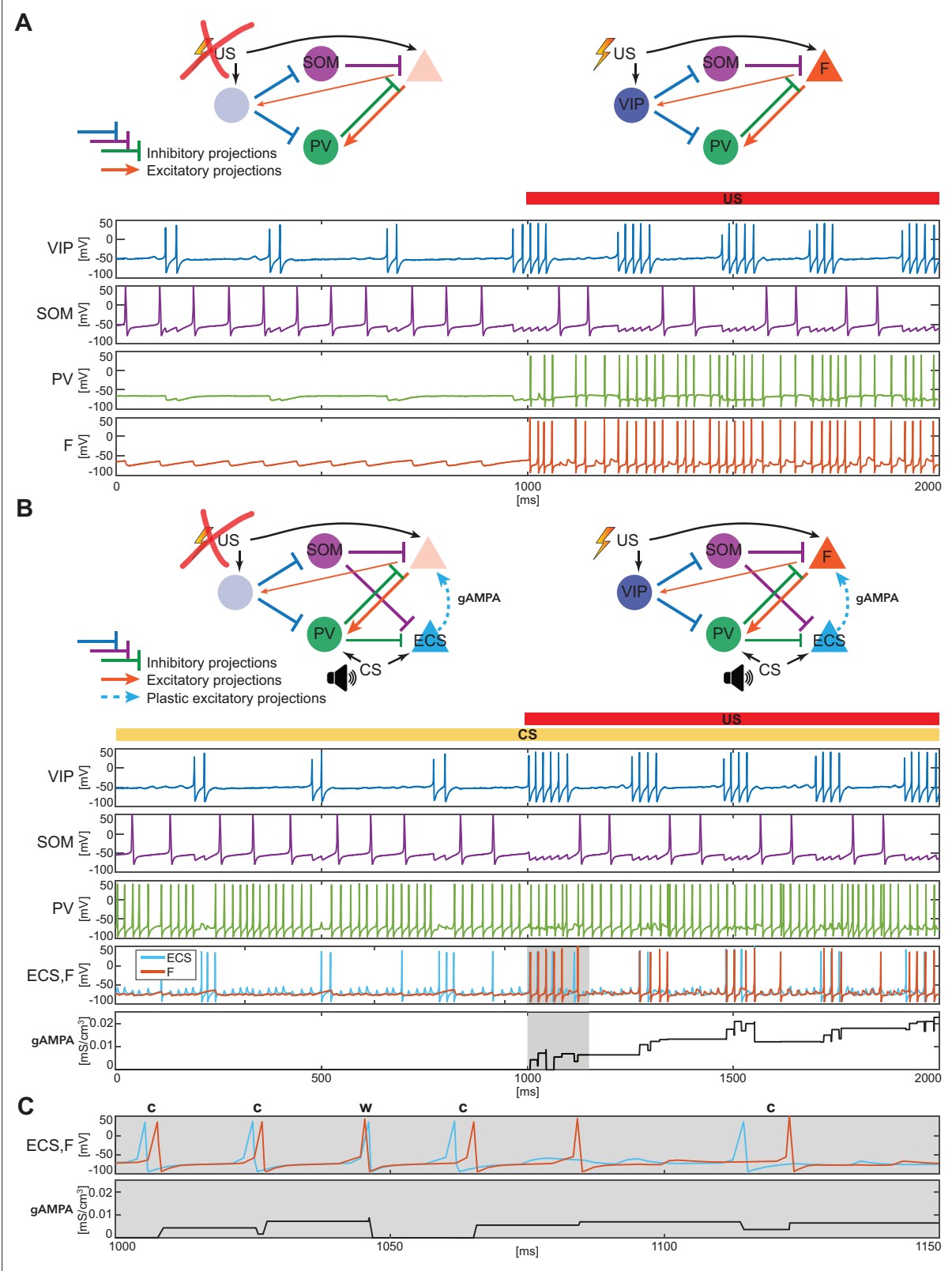

**Figure 2.** BLA interneurons and the excitatory projection neurons interact and modulate the network activity. (**A**) (**top**): Network made of three interneurons (VIP, SOM, and PV) and the excitatory projection neuron encoding fear (**F**) without US input (left) and with US input (right). (**A**) (**bottom**): before the onset of US, VIP shows gamma bursts nested in the low theta rhythm (blue trace), and SOM fires at a natural frequency in the high theta range (purple trace). PV is completely silent due to the lack of any external input (green trace). (**F**) despite its natural frequency of around 11 Hz, is silent

*Figure 2 continued on next page*

*Figure 2 continued*

due to the inhibition from SOM (orange trace). After US onset, due to the longer VIP bursts and the US input, F shows a pronounced activity during the VIP active phase and outside when the SOM and PV inhibition fade. PV is active only when excited by F, and then gives inhibitory feedback to F. (**B**) (**top**): Network in panel A with the excitatory projection neuron encoding the CS input (ECS) during the CS presentation (left) and with paired CS and US inputs (right). (**B**) (**bottom**): 2 s dynamics of all the neurons in the BLA network affected by CS, and by US after 1 s has elapsed. As in panel **A**, VIP shows gamma bursting activity nested in the low theta frequency range with bursts duration affected by the presence or absence of US. VIP inhibits (i) SOM, which fires at high theta (purple trace) regardless of the external inputs, and (ii) PV, which fires at gamma. ECS (light blue trace) and F are both active when both CS and US are present and VIP is active, producing a gamma nested into a low theta rhythm. The evolution in time of the conductance (gAMPA) shows an overall potentiation over the second half of the dynamics when both ECS and F are active. (**C**): blowup of ECS-F burst of activity and gAMPA dynamic shown in the gray area in panel B (**bottom**); ECS (blue trace) fires most of the time right before F, thus creating the correct pre-post timing conducive for potentiation of the ECS to F conductance. The order of each pair of ECS-F spikes is labeled with 'c' (correct) or 'w' (wrong).

cell type. We introduce additional neurons for each cell type with some heterogeneity in the last two sections of the Results.

*Figure 2A* shows a typical dynamic of the network before and after the US input onset. The network produces all the rhythms originating from the interneurons alone or through their interactions with the excitatory projection neuron (shown in *Figure 1*). Specifically, since VIP is active at low theta during both rest and upon the injection of US, it then modulates F at low theta cycles via SOM and PV. In the baseline condition, the VIP interneuron has short gamma bursts nested in a low theta rhythm. With US onset, VIP increases its burst duration and the frequency of the low theta rhythm. These longer bursts make the SOM cell silent for long periods of each low theta cycle, providing F with windows of disinhibition and contributing to the abrupt increase in the activity of F right after the US onset. Finally, in *Figure 2A*, PV lacks any external input and fires only when excited by F. Due to their reciprocal interactions, PV forms a PING rhythm with F, as depicted in *Figure 1C*.

## Interneuron rhythms provide the fine timing needed for depression-dominated STDP to make the association between CS and fear

We now introduce another excitatory projection neuron (ECS), as shown in *Figure 2B* (top). ECS, unlike F, responds to the neutral stimulus CS, as does PV. Similarly to the US, in the rest of the paper, we will use the words 'CS' as shorthand for 'the effects of CS'. In our simulations, CS is modeled as a Poisson spike train at ~50 Hz, independent of the US input. Thus, we hypothesize that the time structure of the inputs sometimes used for the training (e.g. a series of auditory pips) is not central to the formation of the plasticity in the network. Our CS input describes either the context or the cue in contextual and cued fear conditioning, respectively. For the context, the input may come from the hippocampus or other non-sensory regions, but this does not affect its role as input in the model. By the end of fear conditioning, CS consistently activates the neuron F, thus eliciting the network fear response. This happens because of the formation and strengthening of the synapse from ECS to F by means of synaptic plasticity. We now show how this network, with appropriate connection strengths among neurons, can make the timing of the interneurons confer pre-post timing to ECS and F, which is conducive to spike-timing-dependent plasticity potentiation suggested to be critical for associative aversive learning (*Rogan et al., 1997*; *Nabavi et al., 2014*); in particular, we need feedback inhibition (from PV to F) to be stronger than lateral inhibition (from PV to ECS) to promote ECS firing before F. The Hebbian plasticity rule that we use is characterized by a longer time constant of depression than potentiation (and equal maximal amplitudes) and considers the whole history of ECS and F spiking activity (see Materials and methods and *Appendix 1—figure 2A* for more details).

*Figure 2B* shows an example of the network dynamics with CS present for 2 seconds and US injected after the first second of simulation. ECS is active during the whole 2 s interval. CS also affects PV, which is active most of the time; F is active only in the second half of the dynamics when US is also present. All the rhythms generated by the interneurons are apparent in response to simultaneous effects of the CS-US inputs, and they are generated by the same mechanisms as in the fear-only network (*Figure 2A*). In contrast to the fear-only network, the projection neurons F and ECS are both modulated by the VIP low theta rhythm. This is because the PV increased activity due to CS tends to silence (with the help of SOM) ECS and F during the silent VIP phase at low theta. During the active VIP phase, however, both ECS and F are active, and the simulations show that ECS fires most of the time slightly before F (see *Figure 2B and C* and *Appendix 1—figure 1B*); this fine timing needed for

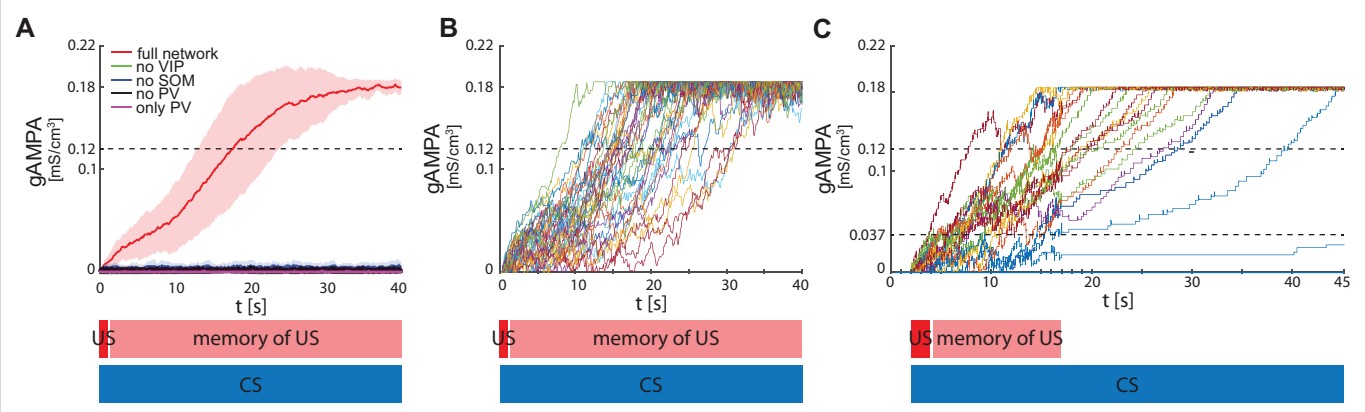

**Figure 3.** ECS to F conductance across network realizations. (**A**) Mean (color-coded curves) and standard deviation (color-coded shaded areas) of the AMPA conductance (gAMPA) from ECS to F across 40 network realizations over 40 s. Red curve and shaded area represent the mean and standard deviation, respectively, across network realizations endowed with all the interneurons. (**B**) Evolution in time of the AMPA conductance for the 40 full network realizations in A. (**C**) AMPA conductance of 20 network realizations over 45 s with F strongly activated by US (2 s) and its effects (13 s), and ECS active the whole interval because of CS; 19 out of 20 network realizations show potentiation after one trial.

potentiation is established by the PING rhythm (see *Figure 1C*). By contrast, in the first second of the simulation in *Figure 2B* (during CS-only), SOM and PV prevent plasticity by silencing F, which in the absence of US receives only a weak applied current.

In the next sections, we will explore the role of each interneuron and its associated rhythm in shaping the network dynamics and allowing the association between CS and fear to be instantiated.

## With the depression-dominated plasticity rule, all interneuron types are needed to provide potentiation during fear learning

We now show that, in the example used above, only the network endowed with all the interneurons and their associated rhythms leads to overall potentiation of the conductance from ECS to F in the timeframe used to induce the fear learning in experimental work. (See Discussion for other plasticity rules.) In general, experimental work finds successful learning after one or very few presentations of CS and US (lasting 1.5 or 2 s) interspersed with CS-only intervals lasting 30–40 s (e.g. see *Davis et al., 2017*; *Krabbe et al., 2019*). The 40 s interval we consider has both ECS and F, as well as VIP and PV interneurons, active during the entire period: an initial bout of US is known to produce a long-lasting fear response beyond the offset of the US (*Hole and Lorens, 1975*) and to induce the release of neuromodulators. The latter, in particular acetylcholine and dopamine that are known to be released upon US presentation (*Harmer and Phillips, 1999*; *Suzuki et al., 2002*; *Rajebhosale et al., 2024*), may induce more sustained activity in the ECS, F, VIP, and PV neurons during and after the presentation of US, thus ensuring a concomitant activation of those neurons necessary for STDP to take place (see 'Assumptions and predictions of the model' in the Discussion).

*Figure 3A* shows the evolution of the average ECS to F AMPA maximal conductance during fear conditioning across 40 network realizations of the full network, as well as from networks lacking VIP, PV, SOM. The average ECS to F AMPA conductance robustly potentiates only in the network containing all three interneurons (*Figure 3A*). Furthermore, as shown in *Figure 3B*, all the full network realizations are 'learners'. We define *learners* as those realizations whose AMPA conductance from ECS to F is higher than 0.12 mS/cm$^3$ at the end of the 40 s interval, which results in the systematic activation of the fear neuron F following most of the ECS spikes when only CS is presented. This is consistent with the high rate of successful learning in rodent experiments after one pairing of CS and US, despite inter-individual differences among animals (*Schafe et al., 2000*). Network realizations differ from one another in the initial state of each neuron involved, and all receive independent Gaussian noise. We note that once the association between CS and fear is acquired, subsequent presentations of CS and US do not weaken or erase it: the interneurons ensure the correct timing and pauses in ECS and F activity, which are conducive for potentiation. Furthermore, since (*Krabbe et al., 2019*) reported that a fraction of PV interneurons are affected by US, we have also run the simulations for single neuron

network with the PV interneuron affected by US instead of CS. In this case as well, all the network realizations are learners (see *Appendix 1—figure 3*).

We show in *Figure 3C* results in which F and VIP show an increased activity due to the US for only 15 seconds of the fear conditioning paradigm, thus relaxing the assumption of these neurons being active for the entire 40 seconds. We find that learning may still occur under this condition. The reason is that, after 15 seconds of enhanced network activity due to CS and US, ECS to F may have potentiated enough that ECS can drive F some of the time (although not all the time). This allows further potentiation to occur in the presence of CS alone. We find that, if the conductance from ECS to F is higher than a threshold value of 0.037 mS/cm³ after 15 s, then the network will become a learner after a further 30 seconds of CS alone; those network realizations that did not reach the threshold value are defined here as *non-learners*. Even with this more restrictive assumption, the large majority of network realizations were learners (19 out of 20 networks). This is in agreement with the experimental fear conditioning literature showing that most of the subjects learn the association between CS and fear after only one trial (*Schafe et al., 2000*).

## Mechanisms by which interneurons contribute to potentiation during depression-dominated plasticity

The PV cell is necessary to induce the correct pre-post timing between ECS and F needed for long-term potentiation of the ECS to F conductance. In our model, PV has reciprocal connections with F and provides lateral inhibition to ECS. Since the lateral inhibition is weaker than the feedback inhibition, PV

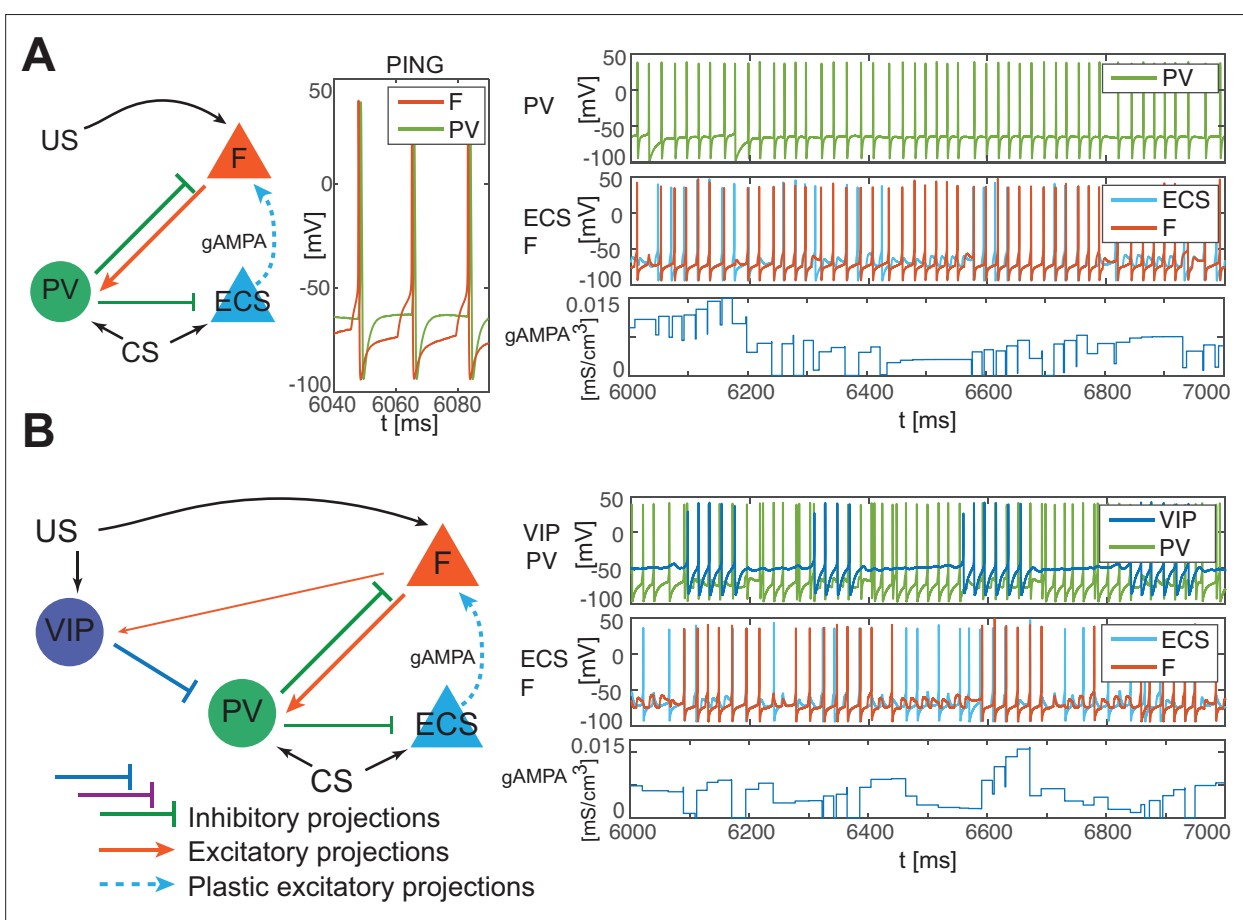

**Figure 4.** PV-only network and VIP-PV network lead to depression. (**A**) Left, network with PV as the only interneuron. PV cell is at an excitation level that supports PING. Middle, dynamics of PV and F that reciprocally interact and generate PING. Right, PV and F entrain in PING (top), ECS and F activity (middle), and ECS to F conductance (bottom). (**B**) Left, network with both PV and VIP. Right, network dynamics (top, middle) followed by the evolution in time of the ECS to F AMPA conductance (bottom). The detailed mechanism behind the evolution of AMPA conductance in panels A and B is in Appendix 1, including *Appendix 1—figure 2*.

tends to bias ECS to fire before F. This creates the fine timing needed for the depression-dominated rule to instantiate plasticity. If we used the classical Hebbian plasticity rule (*Bi and Poo, 2001*) with gamma frequency inputs, this fine timing would not be needed and ECS to F would potentiate over most of the gamma cycle, and thus we would expect random timing between ECS and F to lead to potentiation (*Appendix 1—figure 4*). In this case, no interneurons are needed (See Discussion 'synaptic plasticity in our model' for the potential necessity of the depression-dominated rule).

In the PV and projection cell network configuration, the pre-post timing for ECS and F is repeated robustly over time due to coordinated gamma oscillations (PING, as shown in *Figure 4A*, *Figure 1C*) arising through the reciprocal interactions between F and PV (*Feng et al., 2019*). PING can arise only when PV is in a sufficiently low excitation regime such that F can control PV activity (*Börgers et al., 2005*), as in *Figure 4A*. However, although such a low excitation regime establishes the correct fine timing for potentiation, it is not sufficient to lead to potentiation (*Figure 4A*, *Appendix 1—figure 2C*): the depression-dominated rule leads to depression rather than potentiation unless the PING is periodically interrupted. During the pauses, made possible only in the full network by the presence of VIP and SOM, the history-dependent build-up of depression decays back to baseline, allowing potentiation to occur on the next ECS/F active phase. (The detailed mechanism of how this happens is Appendix 1, including *Appendix 1—figure 2*). Thus, a network without the other interneuron types cannot lead to potentiation. Though a low excitation level for a PV cell is necessary to produce a PING, a higher excitation level is necessary to produce a pause in the ECS and F. This higher excitation level is consistent with the experimental literature showing a strong activation of PV after the onset of CS (*Wolff et al., 2014*). The higher excitation happens when the VIP cell is silent, whereas a low excitation level is achieved when the VIP cell fires and partially inhibits the PV cell (*Figure 4B*, *Appendix 1—figure 2D*). The interruption in the ECS and F activity requires the participation of another interneuron, the SOM cell (*Figure 2B* and *Appendix 1—figure 2*): the pauses in inhibition from the VIP periodically interrupt ECS and F firing by releasing PV and SOM from inhibition and thus indirectly silencing ECS and F. Without these pauses, depression dominates (see Appendix 1 section 'ECS and F activity patterns determine overall potentiation or depression').

## Network with multiple heterogeneous neurons can establish the association between CS and fear

To test the robustness of our single cell results to heterogeneity, we expand our BLA network to include three cells of each interneuron subtype and ten of each excitatory projection neuron (*Figure 5A*). Each neuron has independent noise and cellular parameters (see Materials and methods for details). We find that the network very robustly produces potentiation between the ECS and F affected by CS and US, respectively, during fear training.

*Figure 5B* shows an example of the network dynamics during fear conditioning. As previously presented for the single neuron network (*Figures 2–3*), interneurons are crucial in conferring the correct pre-post spike timing to ECS and F. We assume all the VIP interneurons receive the same US; hence, the VIP neurons tend to approximately synchronize at low theta, allowing a coordinated disinhibition window for potentiation of ECS to F conductance (*Figure 5B*), as we have seen in the single cell model. The potentiation from ECS to F is specifically for the excitatory projection cells affected by CS and US, respectively (*Figure 5C*). The ECS cells not receiving CS are inhibited by ongoing PV activity during the disinhibition window (*Figure 5B*); they are constructed to be firing at 11 Hz in the absence of any connections from other cells. The lack of activity in those cells during fear conditioning implies that there is no plasticity from those ECS cells to the active F. Those cells are included for the calculation of the LFP (see below in 'Increased low theta frequency is a biomarker of fear learning'). This larger network corroborates the results obtained for the single neuron network: only the realizations of the full network learn the association between CS and fear (*Figure 5D*, left) and all those network realizations become learners in less than 40 seconds (*Figure 5D*, right). Similarly, there is a striking failure of plasticity if any interneuron type is removed from the network; even partial plasticity does not arise.

## Increased low theta frequency is a biomarker of fear learning

We find that fear conditioning leads to an increase in low theta frequency power of the network spiking activity compared to the pre-conditioned level (*Figure 6A and B*); there is no change in the

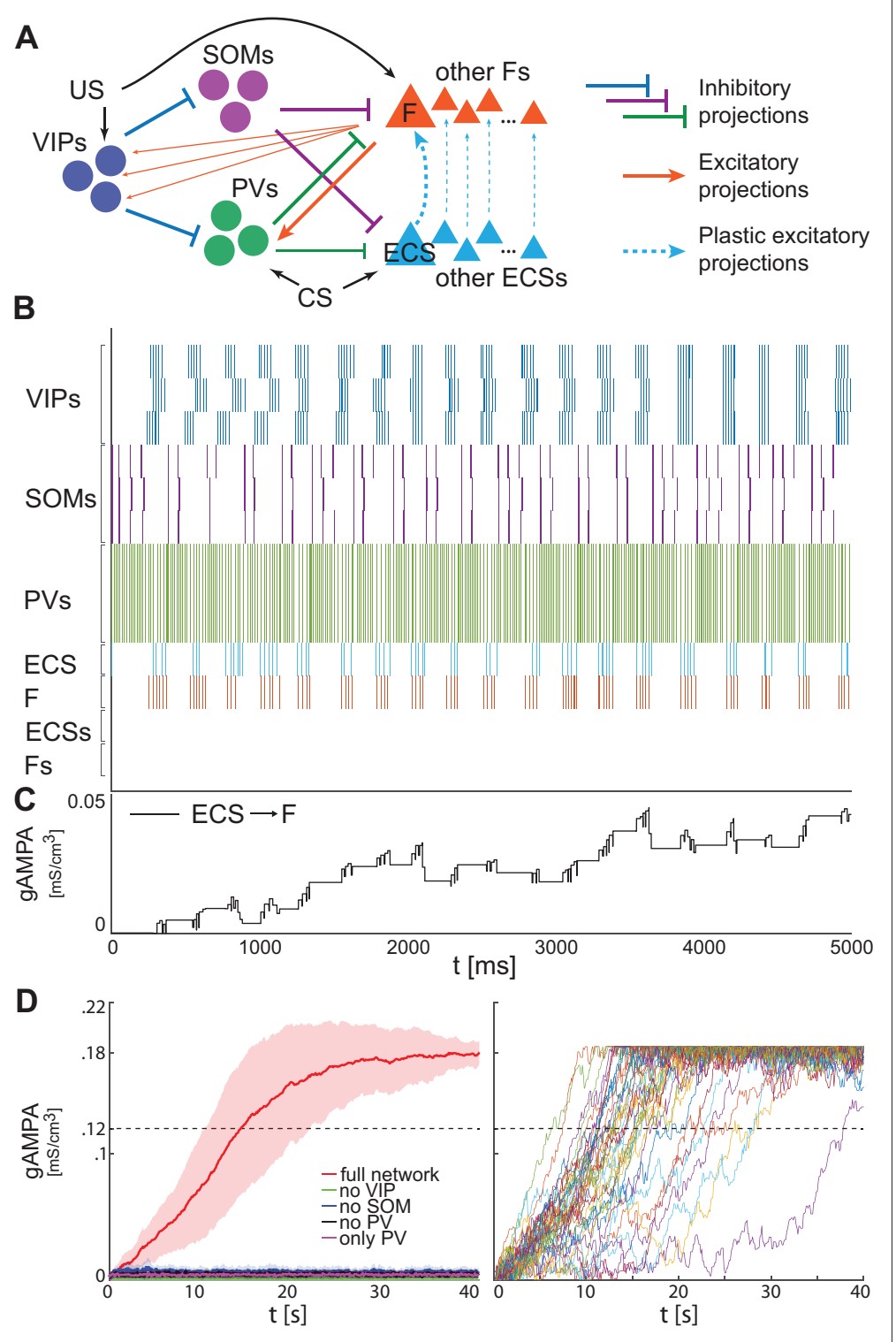

**Figure 5.** Heterogeneous BLA fear network is capable of establishing the association between CS and fear.
(**A**) whole BLA network with multiple and heterogeneous neurons. (**B**) Dynamics in the first 5000ms after the onset of US and CS of each of the neurons in the BLA network. (**C**) Dynamics of ECS to F conductance over 5000 ms shaped by the activity in B. (**D**) Left, mean and standard deviation across 40 network realizations of the ECS to F conductance for the full (red), no VIP (green), no SOM (purple), no PV (black), no SOM and PV (magenta) networks. The green, purple, back, and magenta curves are superimposed on each other. Right, dynamics of all the 40 full network realizations over 40 s.

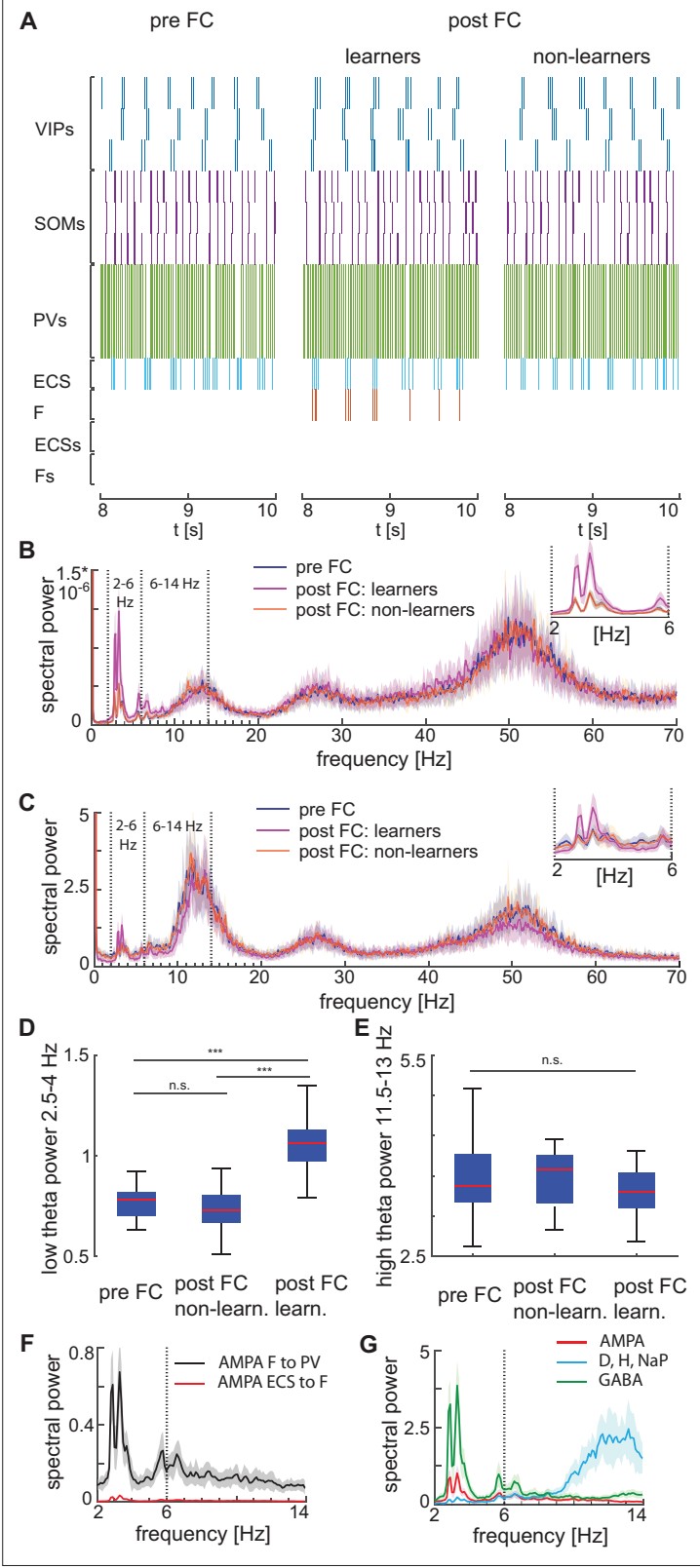

**Figure 6.** Heterogeneous network dynamics and spectral properties pre versus post fear conditioning for network realizations in learners and non-learners. All power spectra are represented as mean and standard deviation across 20 network realizations. (**A**) Dynamic of BLA heterogeneous networks pre (left) and post (learner, middle; non-learner, right) fear conditioning. (**B**) Power spectra of network spiking activity before fear conditioning (blue) and

*Figure 6 continued on next page*

*Figure 6 continued*

after successful (purple) and non-successful fear conditioning (orange); top, right: inset between 2 and 6 Hz. Blue and orange curves closely overlap. (**C**) Power spectra of the LFP proxy (linear sum of AMPA, GABA, D-, NaP-, and H-currents); all the details as in B. (**D, E**) 25th, 50th, and 75th percentiles of LFP low theta power in the 2.5–4 Hz range where the peaks of power exist (**D**) and high theta power in the 12–14 Hz range again where the peaks of power exist (**E**) in 20 network realizations before and after (in both learners and non-learners) fear conditioning. ***: p-value <0.001; n.s.: non-significant difference, obtained using a two-sided Wilcoxon rank sum test (ranksum in Matlab). (**F**) Power spectra of AMPA currents from ECS to F (red curve) and from F to PV interneurons (black curve). (**G**) Power spectra mean and standard deviation of the LFP signals derived from only AMPA currents (red curve), GABA currents (green curve), D-current, NaP-current and H-current (light blue curve). AMPA currents are generated by the interactions from ECS to F, F to VIPs, and F to PVs. VIP cells contribute to the D-current and SOM cells to H-current and NaP-current (see the Result section 'Rhythms in the BLA can be produced by interneurons' for a description of these currents).

---

high theta power. We also find that the LFP, modeled as the linear sum of all the AMPA, GABA, NaP-, D-, and H- currents in the network, similarly reveals a low theta power increase when considering the peak of the low theta power, and no significant variation in the high theta power again when considering the peak of the high theta power (*Figure 6C, D and E*). These results are consistent with the experimental findings in *Davis et al., 2017*. Specifically, the newly potentiated AMPA synapse from ECS to F ensures F is active after fear conditioning, thus generating strong currents in the PV cells to which it has strong connections (*Figure 6F*). It is the AMPA currents to the PV interneurons that are directly responsible for the low theta increase; it is the newly potentiated ECS to F synapse that paces the AMPA currents in the PV interneurons to go at low theta. Thus, the low theta increase is due to added excitation provided by the new learned pathway.

We find that the AMPA currents are the major contributor to the low theta increase. Although both the AMPA and GABA currents contribute to the power increase in the low theta frequency range (*Figure 6G*), the AMPA currents show a dramatic power increase relative to the baseline (the average power ratio of AMPA and GABA post- vs pre-conditioning across 20 network realizations is $3*10^3$ and 4.6, respectively). Thus, the AMPA currents are the major contributors to the low theta power increase. As a further constrain, the additional unresponsive ECS and F cells in the network were included to ensure we had not biased the LFP towards excitation. Finally, the increase in power is in the low theta range because ECS and F are allowed to spike only during the active phase of the low theta spiking VIP neurons. We have also explored another proxy for the LFP (see Appendix 1 and *Appendix 1—figure 5*).

Although the experimental results in *Davis et al., 2017* show an increase in low theta after fear learning, they excluded non-learners from their analysis, and thus, it is unclear from their results if low theta can be used as a biomarker of fear learning. To address this question, we looked at the power spectra after fear conditioning in learning versus non-learning networks. To have an adequate number of non-learners, we ran 60 network simulations for 10 s and chose 20 of them whose conductance from ECS to F remained lower than 0.037 mS/cm$^3$ (i.e. *non-learners*). Notably, the low theta power increase is completely absent after fear conditioning in those network realizations that display no signs of learning (*Figure 6B, C, D*). This suggests that the low theta power change is a biomarker of successful fear conditioning: it occurs when there is learning and does not occur when there is no learning.

## Discussion

### Overview

Our study suggests that amygdalar rhythms play a crucial role in plasticity during fear conditioning. Prominent rhythms found in the BLA during fear conditioning include low theta (~3–6 Hz), high theta (~6–12 Hz), and gamma (>30 Hz) (*Seidenbecher et al., 2003*; *Courtin et al., 2014b*; *Stujenske et al., 2014*; *Davis et al., 2017*). Experimental work in rodents shows that the BLA undergoes more oscillatory firing at low theta frequency and gamma following fear conditioning, while high theta frequency remains unchanged compared to before fear conditioning (*Courtin et al., 2014b*; *Davis et al., 2017*).

To examine the origin of these rhythmic changes and their functional role in fear conditioning, we implement a biophysically detailed model of the BLA. We show that VIP interneurons, PV

interneurons, and SOM interneurons in the BLA may be centrally involved in producing the experimentally measured rhythms based on the biophysical properties of these interneurons. More specifically: the gamma oscillation is associated with PV cell interaction with excitatory cells to produce PING; VIP cells produce low theta due to their intrinsic D-current; SOM cells help produce the high theta rhythm due to persistent sodium current (NaP-current) and H-currents. Moreover, we show that the rhythmic dynamics produced by VIP and PV cells both play a crucial role in the instantiation of plasticity during associative learning by promoting the formation of a dedicated fear circuit shaped through spike-timing-dependence, and the removal of any of the interneuron types from the circuit leads to failure of the plasticity needed for associative fear learning. We note that the presence of SOM cells is crucial for plasticity in our model since they help to produce the necessary pauses in the excitatory projection cell activity. The BLA SOM cells do not necessarily have to be the only source of the high theta observed in the BLA during fear learning; the high theta detected in the LFP of the BLA also originates from the prefrontal cortex and/or the hippocampus (*Stujenske et al., 2014*; *Stujenske et al., 2022*). Finally, we replicate the experimental increase in the low theta after successful training (*Davis et al., 2017*) and determine that this is a biomarker of fear learning, that is the increase in low theta power does not appear in the absence of learning.

## Synaptic plasticity in our model

Synaptic plasticity is the mechanism underlying the association between neurons that respond to the neutral stimulus CS (ECS) and those that respond to fear (F), which instantiates the acquisition and expression of fear behavior. One form of experimentally observed long-term synaptic plasticity is spike-timing-dependent plasticity (STDP), which defines the amount of potentiation and depression for each pair of pre- and postsynaptic neuron spikes as a function of their relative timing (*Bi and Poo, 2001*; *Caporale and Dan, 2008*). All forms of STDP require that there be an overlap in the firing of the pre- and post-synaptic cells. In some fear learning paradigms, the US and the CS do not overlap. We address this below under 'Assumptions and predictions of the model', showing how the effects of US and CS on the spiking of the relevant neurons can overlap even in the absence of overlap of US and CS.

There are many STDP rules in the literature (*Abbott and Nelson, 2000*; *Feldman, 2012*) but none we are aware of specifically for the amygdala. With the depression-dominated rule used in this study, depression is overall stronger than potentiation, unless there is fine timing among pyramidal cells in the gamma and low theta cycles. With this rule, each interneuron plays a role in setting up this fine timing in the gamma and low theta cycles needed for appropriate plasticity. PV interneurons are important in creating a gamma rhythm with F when the latter is activated by the US. This gamma rhythm plays a central role in the fine timing between ECS and F. As mentioned in the Result section 'Mechanisms by which interneurons contribute to potentiation during depression-dominated plasticity', the critical requirements for plasticity are a pause of VIP firing within each low theta cycle and fine timing of the ECS (pre) and F (post) neurons in the active phase of the cycle. The pause in VIP cell firing allows for activation of SOM cells, which helps to inhibit the ECS and F cells. Without the pause, the ECS and F cells continue to fire. It is a consequence of the depression-dominated rule that this would lead to depression; this happens because the depression does not relax to zero between gamma-frequency spikes, and hence continues to build up. With a Hebbian plasticity rule characterized by a lower amplitude for depression than for potentiation, potentiation would occur at most of the phases of gamma and thus fine timing would not be needed (*Appendix 1—figure 4*). The depression-dominated rule allows for regulation of the plasticity by modulation of all the kinds of cells that are known to be involved in fear learning as well as providing reasons for the known involvement of rhythms.

With the classical Hebbian plasticity rule, we suggest that learning can occur without the involvement of the VIP and SOM cells. Although fear learning can occur without the depression-dominated rule, we hypothesize that the latter is necessary for other aspects of fear learning and regulation. Generalization of learning can be pathological, and we hypothesize that the modulation created by the involvement of VIP and SOM interneurons is normally used to prevent such overgeneralization. However, in some circumstances, it may be desirable to account for many possible threats, and then a classical Hebbian plasticity rule could be useful. We note that the involvement or not of the VIP-SOM circuit has been implicated when there are multiple strategies for solving a task (*Piet et al., 2024*). In

our situation, the nature of the task (including reward structure) may determine whether the learning rule is depression-dominated and therefore whether the VIP-SOM circuit plays an important role.

VIP cells may be important in establishing the kind of depression-dominated rule we are using in this model for the synapse between ECS and F. VIP cells are known to corelease the peptide VIP along with GABA (*Bayraktar et al., 1997*). The amount of the release is related to the amount of high frequency firing, that is the duty cycle of the gamma burst in each low theta cycle (*Agoston et al., 1988*; *Agoston and Lisziewicz, 1989*). The peptide VIP can act on second messenger pathways to inhibit potentiation; the pathways are complex and not fully understood, but in hippocampus they involve GABA transmission, NMDA activation, and CaMKII (*Caulino-Rocha et al., 2022*). The relevant VIP receptor VPAC1 is known to exist in the amygdala (*Joo et al., 2005*; *Boucher et al., 2021*). Thus, by inhibiting potentiation, VIP may change a Hebbian plasticity rule to a depression-dominated rule. (See *Cunha-Reis and Caulino-Rocha, 2020* for details about VIP and plasticity). Since high theta is increased during fear extinction, but not learning, and our model suggests an amygdalar source of high theta is the SOM cells, we hypothesize that SST peptide may play some role in aiding the formation of networks involved in fear extinction. Also, the CCK peptide has been proposed to promote the switch between fear and safety states after fear extinction (*Krabbe et al., 2018*).

## Involvement of other brain structures

Studies using fear conditioning as a model of associative learning reveal that learning and expression of fear are not limited to the amygdala but involve a distributed network including the amygdala, the medial prefrontal cortex, and the hippocampus (*Seidenbecher et al., 2003*; *Bocchio and Capogna, 2014*; *Courtin et al., 2014a*; *Stujenske et al., 2014*; *Tovote et al., 2015*; *Karalis et al., 2016*; *Chen et al., 2021*). In our model, the excitatory projection neurons and VIP and PV interneurons show sustained activity during and after the US presentation, thus allowing potentiation through STDP to take place. The medial prefrontal cortex and/or the hippocampus may provide the substrates for the continued firing of the BLA neurons after the 2 s US stimulation. We also discuss below that this network sustained activity may originate from neuromodulator release induced by US (see section 'Assumptions and predictions of the model' in the Discussion).

Other brain structures may be involved in other aspects of fear responsiveness, such as fear extinction and prevention of generalization. It has been reported that the prelimbic cortex (PL) modulates the BLA SOM cells during fear retrieval, and the latter cells are crucial to discriminate non-threatening cues when desynchronized by the PL inputs (*Stujenske et al., 2022*). Also, brain structures such as the prefrontal cortex and hippocampus have been documented to play a crucial role in fear extinction, the paradigm following fear conditioning aimed at decrementing the conditioned fearful response through repeated presentations of the CS alone. As reported by several studies, fear extinction suppresses the fear memory through the acquisition of a distinct memory, instead of through the erasure of the fear memory itself (*Harris et al., 2000*; *Bouton, 2002*; *Trouche et al., 2013*; *Thompson et al., 2018*). *Davis et al., 2017* found a high theta rhythm following fear extinction that was associated with the suppression of threat in rodents. Our model can be extended to include structures in the prefrontal cortex and the hippocampus to further investigate the role of rhythms in the context of discrimination of non-threatening cues and extinction. We hypothesize that a different population of PV interneurons plays a crucial role in mediating competition between fearful memories, associated with a low theta rhythm, and safety memories, associated with a high theta rhythm; supporting experimental evidence is in *Lucas et al., 2016*; *Davis et al., 2017*; *Chen et al., 2022*.

## Where the rhythms originate, and by what mechanisms

A recent experimental paper (*Antonoudiou et al., 2022*) suggests that the BLA can intrinsically generate theta oscillations (3–12 Hz) detectable by LFP recordings when inhibition is totally removed due to gabazine application. They draw this conclusion in mice by removing the hippocampus, which can volume conduct to BLA, and noticing that other nearby brain structures did not display any oscillatory activity. In our model, we note that when inhibition is removed, both AMPA and intrinsic currents contribute to the network dynamics and the LFP. Thus, interneurons with their specific intrinsic currents (i.e. D-current in the VIP interneurons, and NaP- and H- currents in SOM interneurons) can indeed affect the model LFP and support the generation of theta and gamma rhythms (*Figure 6G*).

Another slice study, (*Bratsch-Prince et al., 2024*), shows that BLA is intrinsically capable of producing a low theta rhythm with ACh stimulation and without needing external glutamate input. ACh is produced in vivo by the basal forebrain in response to US (*Rajebhosale et al., 2024*). Although we did not explicitly include the BF and ACh modulation of BLA in our model, we implicitly include the effect of ACh in BLA by increasing the activity of the VIP cells, which then produce the low theta rhythm. Indeed, low theta in the BLA is known to depend on the muscarinic activation of CCK interneurons, a group of interneurons that overlaps with the class of VIP neurons in our model (*Mascagni and McDonald, 2003*; *Krabbe et al., 2018*).

Although the BLA can produce these rhythms, this does not rule out that other brain structures also produce the same rhythms through different mechanisms, and these can be transmitted to the BLA. Specifically, it is known that the olfactory bulb produces and transmits the respiratory-related low theta (4 Hz) oscillations to the dorsomedial prefrontal cortex, where it organizes neural activity (*Bagur et al., 2021*). Thus, the respiratory-related low theta may be captured by BLA LFP because of volume conduction or through BLA extensive communications with the prefrontal cortex. Furthermore, high theta oscillations are known to be produced by the hippocampus during various brain functions and behavioral states, including during spatial exploration (*Vanderwolf, 1969*) and memory formation/retrieval (*Raghavachari et al., 2001*), which are both involved in fear conditioning. Similarly to the low theta rhythm, the hippocampal high theta can manifest in the BLA. It remains to understand how these other rhythms may interact with the ones described in our paper. However, we emphasize that there is also evidence (as discussed above) that these rhythms arise within the BLA.

## Assumptions and predictions of the model

The interneuron descriptions in the model were constrained by the electrophysiological properties reported in response to hyperpolarizing currents (*Sosulina et al., 2010*). Specifically, we modeled the three subtypes of VIP, SOM, and PV interneurons displaying bursting behavior, regular spiking with early spike-frequency adaptation, and regular spiking without spike-frequency adaptation, respectively. Focusing on VIP interneurons, we were able to model the bursting behavior by including the D-type potassium current. This current is thought to exist in the VIP interneurons in the cortex (*Porter et al., 1998*), but whether this current is also found in the BLA is still unknown. We modeled the SOM interneurons with NaP- and H-currents, similar to the OLM cells in the hippocampus. Due to these currents, the VIP and SOM cells are able to produce low- and high theta oscillations, respectively. The presence of these currents, and the neurons' ability to exhibit oscillations in the theta range during fear conditioning and at baseline in BLA, which are assumptions of our model, should be tested experimentally. Our model predicts that blockade of D-current in VIP interneurons (or silencing VIP interneurons) will both diminish low theta and prevent fear learning. Finally, the model assumes the absence of significantly strong connections from the excitatory projection cells ECS to PV interneurons, unlike the ones from F to PV. Including those synapses would alter the PING rhythm created by the interactions between F and PV, which is crucial for fine timing between ECS and F needed for LTP.

Our model, which is a first effort towards a biophysically detailed description of the BLA rhythms and their functions, does not include the neuron morphology, many other cell types, conductances, and connections that are known to exist in the BLA; models such as ours are often called 'minimal models' and constitute most biologically detailed models. For example, although there is considerable variability in the activity patterns of both VIP cells and SOM cells (*Sosulina et al., 2010*; *Guthman et al., 2020*; *Ünal et al., 2020*; *Vereczki et al., 2021*), our focus was specifically on those subtypes that generate critical rhythms within the BLA. Such minimal models are used to maximize the insight that can be gained by omitting details whose influence on the answers to the questions addressed in the model are believed not to be qualitatively important. We note that the absence of these omitted features constitutes hypotheses of the model: we hypothesize that the absence of these features does not materially affect the conclusions of the model about the questions we are investigating. Of course, such hypotheses can be refuted by further work showing the importance of some omitted features for these questions and may be critical for other questions. Our results hold when there is some degree of heterogeneity of cells of the same type, showing that homogeneity is not a necessary condition.

Our study suggests that all the interneurons modeled are necessary for associative learning provided that the STDP rule is depression-dominated. This prediction could be tested experimentally by selectively silencing each interneuron subtype in the BLA: if the associative learning is hampered by

silencing any of the interneuron subtypes, this supports our study. Finally, the model prediction could be tested indirectly by acquiring more information about the plasticity rule involved in the BLA during associative learning. We found that all the interneurons are necessary to establish fear learning only in the case of a depression-dominated rule. This rule ensures that fine timing and pauses are always required for potentiation: interneurons provide both fine timing and pauses to pyramidal cells, making them crucial components of the fear circuit.

Finally, our model requires the effect of the CS and US inputs on the BLA neuron activity to overlap in time in order to instantiate fear learning through STDP. Such a hypothesis, that learning uses spike-timing-dependent plasticity, is common in the modeling literature (*Bi and Poo, 2001*; *Caporale and Dan, 2008*; *Markram et al., 2011*). Current paradigms of fear conditioning include examples in which the CS and US stimuli do not overlap (*Krabbe et al., 2019*). Such a condition might seem to rule out the mechanisms in our paper. Nevertheless, the argument below suggests that the effects of the CS and US can cause an overlap in neuronal spiking of ECS, F, VIP, and SOM, even when CS and US inputs do not overlap.

Experimental recordings cannot speak to the rate of spiking of BLA neurons during US due to recording interference from the shock. However, evidence suggests that ECS activity should increase during the US due to the release of acetylcholine (ACh) from neurons in the basal forebrain (BF) (*Rajebhosale et al., 2024*). Pyramidal cells of the BLA robustly express M1 muscarinic ACh receptors (*McDonald and Mott, 2021*). Thus, ACh from BF should elicit a depolarization in pyramidal cells. Indeed, the pairing of ACh with even low levels of spiking of BLA neurons results in a membrane depolarization that can last 7–10 s (*Unal et al., 2015*). Other modulators, including dopamine, may also play a role in producing the sustained activity. Activation of US leads to increased dopamine release in the BLA (*Harmer and Phillips, 1999*; *Suzuki et al., 2002*). D1 receptors are known to increase the membrane excitability of BLA projection neurons by lowering their spiking threshold (*Kröner et al., 2005*). Thus, neuromodulator release should induce higher spiking rates and more sustained activity in the ECS and F neurons during and after the presentation of US, thus ensuring a concomitant activation of ECS and fear (F) neurons necessary for STDP to take place. Thus, the activation of the US can lead to continued and higher firing rates of ECS and F. The effect of dopamine can last up to 20 minutes (*Kröner et al., 2005*). For CS-positive neurons, the ACh modulation coming from the firing of US may lead to a temporary extension of firing that is then amplified and continued by dopaminergic effects.

Hence, we suggest that a solution to the problem apparently posed by the non-overlap US and CS in some paradigms of auditory fear conditioning (*Krabbe et al., 2019*) may be solved by considering the roles of ACh and dopamine in the BLA. The model we have may be considered a "minimal" model that puts in by hand the overlap in activity due to the neuromodulation without explicitly modeling it. We have used the simplest way to model overlap without assumptions about timing specificity in the overlap. We note that, even though ECS and F neurons have the ability to fire continuously when ACh and dopamine are involved, the participation of the interneurons enforces periodic silence needed for the depression-dominated STDP.

## Is STDP needed in fear conditioning?

The study in *Grewe et al., 2017* questions the validity of the Hebbian model in establishing associative learning during fear conditioning. There are several critiques we discuss here. The first critique is that Hebbian plasticity does not explain the experimental finding showing that both upregulation and downregulation of stimulus-evoked responses are present between coactive neurons. The upregulation is provided by our model, so the issue is the downregulation, which is not addressed by our model. However, our model highlights that coactivity alone does not create potentiation; the fine timing of the pre- and postsynaptic spikes determines whether there is potentiation or depression. Here, we find that PING networks are instrumental in setting up the fine timing for potentiation. We suggest that networks not connected to produce the PING may undergo depression when coactive.

The second critique raised by *Grewe et al., 2017* is that Hebbian plasticity alone does not explain why most of the cells exhibiting enhanced responses to the CS did not react to the US before fear conditioning. They suggest that neuromodulators may provide a third condition (besides the activity of the pre- and postsynaptic neurons) that changes the plasticity rule. Our model also does not explicitly address this experimental finding since it requires F to be initially activated by US in order for the fear association to be established. We agree that the fear cells described in *Grewe et al., 2017* may

be depolarized by the US without reaching the spiking threshold; however, with neuromodulation provided during the fear training, the same input can lead to spiking, enabling the conditions for Hebbian plasticity. Our discussions above about how neuromodulators affect excitability are relevant to this point. We do not exclude that other forms of plasticity may play a role during fear conditioning in cells not initially activated by the US, but this is not the topic of our modeling study.

The third critique raised by *Grewe et al., 2017* is that Hebbian plasticity cannot explain why the majority of cells that were US- and CS-responsive before training have a reduced CS-evoked response afterward. The reduced response happens over multiple exposures of CS without US; this can involve processes similar to those present in fear extinction, which require plasticity in further networks, especially involving the infralimbic cortex (*Milad and Quirk, 2002*; *Burgos-Robles et al., 2007*). An extension of our model could investigate such mechanisms. In the fourth critique, *Grewe et al., 2017* suggests that the Hebbian plasticity rule cannot easily account for the reduction of the responses of many CS+-responsive cells, but not of the CS−-responsive cells. We suggest that the circuits involving paradigms similar to fear extinction do not involve the CS- cells.

Overall, we agree with (*Grewe et al., 2017*) that neuromodulators play a crucial role in fear conditioning, especially in prolonging the US- and CS-encoding activity as discussed in (see section 'Assumptions and predictions of the model' in the Discussion), or even participating in changing the details of the plasticity rule. A possible follow-up of our work involves investigating how fear ensembles form and modify through fear conditioning and later stages. This follow-up work may involve using a tri-conditional rule, as suggested in *Grewe et al., 2017*, in which the potential role of neuromodulators is taken into account in the plasticity rule in addition to the pre- and postsynaptic neuron activity. Another direction is to investigate a possible relationship between neuromodulation and a depression-dominated Hebbian rule.

## Comparison with other models

Many computational models that study fear conditioning have been proposed in the last years; the list includes biophysically detailed models (e.g., *Li et al., 2009*; *Kim et al., 2013*), firing rate models (e.g. *Krasne et al., 2011*; *Vlachos et al., 2011*; *Ball et al., 2012*), and connectionist models (e.g. *Edeline and Weinberger, 1992*; *Armony et al., 1997*; *Moustafa et al., 2013*; for a review see *Nair et al., 2016*). Both firing rate models and connectionist models use an abstract description of the interacting neurons or regions. The omission of biophysical details prevents such models from addressing questions concerning the roles of dynamics and biophysical details in fear conditioning, which is the aim of our model. There are also biophysically detailed models (*Li et al., 2009*; *Kim et al., 2013*; *Kim et al., 2016*; *Feng et al., 2019*), which differ from ours in both the physiology included in the model and the description of how plastic changes take place. One main difference in the physiology is that we differentiated among types of interneurons, since the fine timing produced for the latter was key to our use of rhythms to produce spike-timing-dependent plasticity. The origin of the gamma rhythm (but not the other rhythms) was investigated in *Feng et al., 2019*, but none of these papers connected the rhythms to plasticity.

The most interesting difference between our work and that in *Li et al., 2009*; *Kim et al., 2013*; *Kim et al., 2016* is the modeling of plasticity. We use spike-timing-dependent plasticity rules. The models in *Li et al., 2009*; *Kim et al., 2013*; *Kim et al., 2016* were more mechanistic about how the plasticity takes place, starting with the known involvement of calcium with plasticity. Using a hypothesis about back propagation of spikes, the set of papers together develop a theory that is consistent with STDP and other instantiations of plasticity (*Shouval et al., 2002a*). For the purposes of our paper, this level of detail, though very interesting, was not necessary for our conclusions. By contrast, in order for the rhythms and the interneurons to have the dynamic roles they play in the model, we needed to restrict our STDP rule to ones that are depression-dominated. Our reading of *Shouval et al., 2002b* suggests to us that such subrules are possible outcomes of the general theory. Thus, there is no contradiction between the models, just a difference in focus; our focus was on the importance of the much-documented rhythms (*Seidenbecher et al., 2003*; *Courtin et al., 2014a*; *Stujenske et al., 2014*; *Davis et al., 2017*) in providing the correct spike timing. We showed in Appendix 1 ("Classical Hebbian plasticity rule, unlike the depression-dominated one, shows potentiation even with no strict pre and postsynaptic spike timing") that if the STDP rule was not depression dominated, the rhythms need not be necessary. We hypothesize that the necessity of strict timing enforced by the

depression-dominated rule may foster the most appropriate association with fear at the expense of less relevant associations.

The dynamics of the VIP cell play a central role in the plasticity we investigate. This is in contrast to the cortical model in *Veit et al., 2023* for which VIP is essential for locally controlling gain and globally controlling coherence in gamma oscillations. In the model by *Veit et al., 2023*, the global control requires differences in long-range connectivity that are known to exist and are inserted in the model by hypothesis. Our paper shows how more detailed biophysics produces rhythms among the interneurons used in *Veit et al., 2023* and how these rhythms can produce the plasticity needed to construct those differences in long-range connectivity. Thus, although *Veit et al., 2023* shows that rhythms are not needed for some kinds of control once connectivity is established, our paper suggests that the same set of interneurons, with more detailed physiology, can support the establishment of appropriate connectivity as well as the control described in *Veit et al., 2023*. We note that *Veit et al., 2023* deal with cortical networks, while our model describes BLA networks; however, it is known that these networks are structurally related (*Sah et al., 2003*; *Tovote et al., 2015*; *Polepalli et al., 2020*).

## Limitations and caveats

LFPs recorded in the experiments are thought to be mainly created by transmembrane currents in neurons located around the electrode and depend on several factors, including the morphology of the arborization of contributing neurons and the location of AMPA and GABA boutons (*Katzner et al., 2009*; *Lindén et al., 2011*; *Łęski et al., 2013*; *Mazzoni et al., 2015*). Since our model has no spatial extension, we used an LFP proxy; this proxy was shown to reflect the rhythmic output of the network, which we believe to be the essential result (for more details see Results 'Increased low theta frequency is a biomarker of fear learning', and Appendix 1 'A higher low theta power increase emerges in LFP approximated with the sum of the absolute values of the currents compared to their linear sum').

The use of small number of neurons raises the issue of model scalability. One way in which large models can be different from much smaller ones is in heterogeneity of the neurons of any given type. By using a network with a few neurons of each type, we have begun the study of effects of heterogeneity. In general, the use of small numbers, which effectively assumes that each cell represents a synchronous subset of a larger population, replaces the use of gap-junctions that are known to exist in the cortex among VIP cells (*Francavilla et al., 2018*) as well as among SOM cells and among PV cells (*Tremblay et al., 2016*).

We do not explicitly model the biophysics of NMDA receptors. Rather, we model the effect of such receptors using the spike-timing-dependent plasticity resulting from such biophysics, as is commonly done when modeling STDP (*Song et al., 2000*). Also, our neurons are single-compartment, so do not build in the spatial structure known to exist on the dendrites (*Blair et al., 2001*; *Bennett et al., 2019*).

Our model assumes that initial stages of fear learning can be accomplished entirely within the amygdala, though it is known that other structures in the brain are important for modulating networks related to fear. Much of the work involving the prefrontal cortex and the hippocampus relates to fear extinction, which is not addressed in this paper.

## Summary

We have shown how networks of amygdala neurons, including multiple types of interneurons, can work together to produce plasticity needed for fear learning. The coordination necessary to produce the plasticity requires the involvement of multiple rhythms. Thus, our paper both accounts for the experimental evidence showing such amygdala rhythms exist and points to their central role in the mechanisms of plasticity involved in associative learning. These mechanisms may be common to other types of associative learning, as similar interneuron subtypes and connectivity are ubiquitous in the cortex (*Sah et al., 2003*; *Tovote et al., 2015*; *Polepalli et al., 2020*).

# Materials and methods
## Neuron model

Our network is made of interacting single-compartment neurons modeled using conductance-based models with Hodgkin-Huxley-type dynamics. The temporal voltage change of each neuron is described by:

$$c_m \frac{dV}{dt} = -\sum I_{membrane} - \sum I_{synaptic} + I_{app} + I_{noise},$$

where, $c_m$ is the membrane capacitance, and $I_{membrane}$ are the intrinsic membrane currents, which include a fast sodium current ($I_{Na}$), a fast potassium current ($I_K$), and a leak current ($I_L$ for all neuron types. VIP interneurons additionally have a D-current, and SOM interneurons additionally have NaP and H-currents (*Rotstein et al., 2005*; *Tort et al., 2007*). All these currents are discussed in more detail below, where we describe each neuron individually. The synaptic currents ($I_{synaptic}$) take into account the input from the other neurons in the network and depend on the network connectivity and specific type of synaptic input, as discussed below. Finally, the background drive $I_{app}$ is a constant term that determines the background excitation of a neuron, and $I_{noise}$ corresponds to a Gaussian noise input with mean zero, standard deviation 1, and a specific amplitude for each neuronal cell type (specified below). All the currents are expressed in units per area, rather than absolute units, to avoid making assumptions about the size of the neuron surface.

## Membrane currents

The membrane currents $I_{Na}$, $I_K$, and $I_L$ are modeled using Hodgkin-Huxley-type conductances formulated as:

$$I_{Na}\left(V, h\right) = \bar{g}_{Na} m_\infty^3 h \left(V - E_{Na}\right)$$
$$\text{or alternatively } I_{Na}\left(V, h\right) = \bar{g}_{Na} m^3 h \left(V - E_{Na}\right)$$
$$I_K\left(V, n\right) = \bar{g}_K n^4 \left(V - E_K\right)$$
$$I_L\left(V\right) = \bar{g}_L \left(V - E_L\right)$$

(1)

Each membrane current has a constant maximal conductance $\bar{g}_{channel}$ and a reversal potential $E_{channel}$ (for *channel* = Na, K, or L). The activation ($m$ and $n$) and inactivation ($h$) gating variables evolve in time according to:

$$\frac{dx}{dt} = \frac{x_\infty - x}{\tau_x},$$

(2)

where $x = m$, $n$, $h$. The steady-state function ($x_\infty$) and the time constant of decay ($\tau_x$), which are taken from previous models (*Mainen and Sejnowski, 1996*; *Olufsen et al., 2003*), are formulated as rate functions for each opening ($\alpha_x$) and closing ($\beta_x$ of the ionic channel through:

$$x_\infty = \frac{\alpha_x}{\alpha_x + \beta_x}$$

$$\tau_x = \frac{1}{\alpha_x + \beta_x}.$$

(3)

The specific functions and constants for each cell type in the network are given below.

### Vasoactive intestinal peptide interneurons (VIP)

The membrane currents ($I_{membrane}$) of the VIP interneurons consist of a fast sodium current ($I_{Na}$) (described as in the first formulation of $I_{Na}$ in *Equation 1*), a fast potassium current ($I_K$), a leak current ($I_L$, as in *Equation 1*, and a potassium D-current [$I_D$]). The formulations of these currents were derived from a previous model of cortical interneurons (*Golomb et al., 2007*) and subsequently used to model striatal fast spiking interneurons (*Sciamanna and Wilson, 2011*; *Chartove et al., 2020*), which are reported below.

The maximal sodium conductance is $-g_{Na} = 112.5\,mS/cm^2$ and the sodium reversal potential is $E_{Na} = 50\,mV$. The steady state functions for the sodium current activation ($m$) and inactivation ($h$) variables and $h$ time constant ($\tau_h$) are described by:

$$m_\infty = \frac{1}{1 + \exp\left[-\left(V + 24\right)/11.5\right]}$$

$$h_\infty = \frac{1}{1 + \exp\left[(V + 58.3)/6.7\right]}$$

$$\tau_h = 0.5 + \frac{14}{1 + \exp\left[(V + 60)/12\right]}.$$

The maximal conductance for the fast potassium channel is $-g_K = 225\,mS/cm^2$ and the potassium reversal potential is $E_K = -90\,mV$. The fast potassium channel has no inactivation gates and two activation gates described as follows:

$$n_\infty = \frac{1}{1 + \exp\left[-(V + 12.4)/6.8\right]}$$

$$\tau_n = \left(0.087 + \frac{11.4}{1 + \exp\left[(V + 14.6)/8.6\right]}\right)\left(0.087 + \frac{11.4}{1 + \exp\left[-(V - 1.3)/18.7\right]}\right).$$

The leak current ($I_L$ has no gating variables. The maximal leak conductance is $-g_L = 0.25\,mS/cm^2$ and the leak channel reversal potential is $E_L = -70\,mV$.

The fast-activating, slowly inactivating potassium D-current $I_D$ is formulated as described in **Golomb et al., 2007**:

$$I_D(V, a, b) = \bar{g}_D a^3 b (V - E_K)$$

$$\frac{da}{dt} = \frac{a_\infty - a}{\tau_a}$$

$$\frac{db}{dt} = \frac{b_\infty - b}{\tau_b},$$

with maximal conductance $\bar{g}_D = 3\,mS/cm^2$. The steady state functions for the activation ($a$) and inactivation ($b$) variables are described as follows:

$$a_\infty = \frac{1}{1 + \exp\left[-(V + 50)/20\right]}$$

$$b_\infty = \frac{1}{1 + \exp\left[(V + 70)/6\right]},$$

while the time constant of the decay is $\tau_\alpha = 2ms$ for the activation gate and $\tau_\beta = 150ms$ for the inactivation gate.

In the absence of US, the applied current $I_{app}$ is set to $4\,\mu A/cm^2$. When US is present, $I_{app} = 5\,\mu A/cm^2$. The Gaussian noise ($I_{noise}$) has mean 0, standard deviation 1, and an amplitude of $5\sqrt{\delta t}$, where $\delta t = 0.05\,ms$ corresponds to the time step of integration in our simulations.

### Somatostatin-positive interneurons (SOM)

The membrane currents ($I_{membrane}$) of the SOM interneurons consist of a fast sodium current ($I_{Na}$) (described as in the second formulation of $I_{Na}$ in **Equation 1**), a fast potassium current ($I_K$), and a leak current ($I_L$ as in **Equation 1**, along with an H-current ($I_H$) and NaP-current ($I_P$). The formulations of these currents were taken from previous models of the oriens lacunosum-moleculare (SOM-positive O-LM) cells in the hippocampus (**Rotstein et al., 2005**; **Tort et al., 2007**) and are reported below.

The maximal sodium conductance is $-g_{Na} = 52\,mS/cm^2$ and the sodium reversal potential is $E_{Na} = 55\,mV$. The rate functions for the sodium current activation ($m$) and inactivation ($h$) variables in **Equations 2-3** are formulated as follows:

$$\alpha_m = \frac{-0.1(V + 23)}{\exp\left[-0.1(V + 23) - 1\right]}$$

$$\beta_m = 4\exp\left[-(V + 48)/18\right]$$

$$\alpha_h = 0.07\exp\left[-(V + 37)/20\right]$$

$$\beta_h = \frac{1}{\exp\left[-0.1\left(V+7\right)+1\right]}.$$

The maximal potassium conductance is $-g_K = 11\,mS/cm^2$ and the potassium reversal potential is $E_K = -90\,mV$. The rate functions for the potassium current activation variable ($n$) are formulated as follows:

$$\alpha_n = \frac{-0.01\left(V+27\right)}{\exp\left[-0.1\left(V+27\right)-1\right]}$$

$$\beta_n = 0.125\exp\left[-\left(V+37\right)/80\right].$$

The leak current ($I_L$ has no gating variables. The maximal leak conductance is $-g_L = 0.62\,mS/cm^2$ and the leak channel reversal potential is $E_L = -65\,mV$.

The slow hyperpolarization-activated mixed cation current $I_H$ is formulated as described in **Rotstein et al., 2005**:

$$I_H\left(V, h^f, h^s\right) = \bar{g}_H\left(0.65h^f + 0.35h^s\right)\left(V - E_H\right)$$

$$\frac{dh^f}{dt} = \frac{h_\infty^f - h^f}{\tau_{hf}}$$

$$\frac{dh^s}{dt} = \frac{h_\infty^s - h^s}{\tau_{hs}},$$

with maximal conductance $\bar{g}_H = 1.45\,mS/cm^2$ and $E_H = -20\,mV$. The steady state functions for the $h^f$ and $h^s$ variables and their time constant of decay are described as follows:

$$h_\infty^f = \frac{1}{1 + \exp\left[\left(V + 79.2\right)/9.78\right]}$$

$$\tau_{hf} = \frac{0.51}{\exp\left[\left(V - 1.7\right)/10\right] + -\exp\left[\left(V + 340\right)/52\right]} + 1$$

$$h_\infty^s = \frac{1}{1 + \exp\left[\left(V + 2.83\right)/15.9\right]^{58}}$$

$$\tau_{hs} = \frac{5.6}{\exp\left[\left(V - 1.7\right)/14\right] + \exp\left[-\left(V + 260\right)/43\right]} + 1$$

The persistent sodium current $I_P$ is formulated as described in **Rotstein et al., 2005**; **Rotstein et al., 2006**:

$$I_P = \bar{g}_P p\left(V - E_{Na}\right).$$

The maximal persistent sodium conductance is $\bar{g}_P = 0.5\,mS/cm^2$ and the sodium reversal potential is, as stated above, $E_{Na} = 55\,mV$. The steady state function for the persistent sodium current $I_P$ ($p_\infty$) and the time constant ($\tau_p$) are described by:

$$p_\infty = \frac{1}{1 + \exp\left[-\left(V + 38\right)/6.5\right]}$$

$$\tau_p = 0.15.$$

Throughout all simulations, the applied current $I_{app}$ is set to $0.1\,\mu A/cm^2$. The Gaussian noise ($I_{noise}$) has mean 0, standard deviation 1, and an amplitude of $4\sqrt{\delta t}$, where $\delta t = 0.05\,ms$ corresponds to the time step of integration in our simulations. We note that the persistent sodium current can be replaced by an A-current to produce a high theta rhythm (**Gloveli et al., 2005**).

## Parvalbumin-positive interneurons (PV)

The membrane currents ($I_{membrane}$) of the PV interneurons consist of only a fast sodium current ($I_{Na}$; described as in the second formulation of $I_{Na}$ in **Equation 1**), a fast potassium current ($I_K$), and a leak current ($I_L$, as in **Equation 1**.

The maximal sodium conductance is $\bar{g}_{Na} = 100\,mS/cm^2$ and the sodium reversal potential is $E_{Na} = 50\,mV$. The rate functions for the sodium current activation ($m$) and inactivation ($h$) variables are formulated as follows:

$$\alpha_m = \frac{0.32\,(V+54)}{1 - \exp\left[-\,(V+54)\,/4\right]}$$

$$\beta_m = \frac{0.28\,(V+27)}{\exp\left[(V+27)\,/5\right] - 1}$$

$$\alpha_h = 0.128\,\exp\left[-\,(V+50)\,/18\right]$$

$$\beta_h = \frac{4}{1 + \exp\left[-\,(V+27)\,/5\right]}.$$

The maximal potassium conductance is $\bar{g}_K = 80\,mS/cm^2$ and the potassium reversal potential is $E_K = -100\,mV$. The rate functions for the potassium current activation ($n$) variables are formulated as follows:

$$\alpha_n = \frac{0.032\,(V+52)}{1 - \exp\left[-\,(V+52)\,/5\right]}$$

$$\beta_n = 0.5\,\exp\left[-\,(V+57)\,/40\right].$$

The leak current ($I_L$ has no gating variables. The maximal leak conductance is $\bar{g}_L = 0.1\,mS/cm^2$ and the leak channel reversal potential is $E_L = -67\,mV$. Throughout all the simulations, $I_{app} = 0\,\mu A/cm^2$. The Gaussian noise ($I_{noise}$) has mean 0, standard deviation 1, and an amplitude of $4\sqrt{\delta t}$, where $\delta t = 0.05\,ms$ corresponds to the time step of integration in our simulations.

## Excitatory projection neurons (ECS and F)

The membrane currents ($I_{membrane}$) of ECS and F consist of a fast sodium current ($I_{Na}$) (described as in the first formulation of $I_{Na}$ in **Equation 1**), a fast potassium current ($I_K$), and a leak current ($I_L$) as in **Equation 1**.

The maximal sodium conductance is $\bar{g}_{Na} = 100\,mS/cm^2$ and the sodium reversal potential is $E_{Na} = 50\,mV$. The rate functions for the sodium current activation ($m$) and inactivation ($h$) variables are formulated as follows:

$$\alpha_m = \frac{0.1\,(V+35)}{1 - \exp\left[-\,(V+35)\,/10\right]}$$

$$\beta_m = 4\exp\left[-\,(V+60)\,/18\right]$$

$$\alpha_h = 0.07\exp\left[-\,(V+58)\,/20\right]$$

$$\beta_h = \frac{1}{\exp\left[-0.1\,(V+28)\right] + 1}.$$

The maximal potassium conductance is $\bar{g}_K = 80\,mS/cm^2$ and the potassium reversal potential is $E_K = -100\,mV$. The rate functions for the potassium current activation ($n$) variables are formulated as follows:

$$\alpha_n = \frac{-0.01\,(V+34)}{\exp\left[-0.1\,(V+34) - 1\right]}$$

$$\beta_n = 0.125\exp\left[-\,(V+44)\,/80\right].$$

The leak current ($I_L$ has no gating variables. The maximal leak conductance is $\bar{g}_L = 0.1\,mS/cm^2$ and the leak channel reversal potential is $E_L = -67\,mV$. The formulations of these currents were taken from the description of excitatory/inhibitory neurons presented in *Zhou et al., 2018*.

When neither US nor CS are injected, the applied current $I_{app,F}$ is set to $0.35\,\mu A/cm^2$ and $I_{app,ECS}$ is set to $0.45\,\mu A/cm^2$. By contrast, $I_{app,F}$ is set to $0.5\,\mu A/cm^2$ when US is injected. For both ECS and F, the Gaussian noise ($I_{noise}$) has mean 0, standard deviation 1, and an amplitude of $4\sqrt{\delta t}$, where $\delta t = 0.05\,ms$ corresponds to the time step of integration in our simulations.

## Conditioned and unconditioned stimuli

The conditioned (CS) and unconditioned (US) stimuli affect specific cell types according to the fear conditioning phase. CS consists of a Poisson spike train ($\lambda$=800) that excites an auxiliary excitatory neuron (described by the same equations used for ECS in the previous section). The auxiliary excitatory neuron excites both ECS and PV and makes them fire, in isolation, at ~50 Hz. The maximal AMPA conductance from the auxiliary excitatory neuron to PV and ECS is $-g_e = 0.2\,mS/cm^2$ (see the next paragraph for a description of the AMPA synapses). Similarly, US ($\lambda$=800, independent of CS) affects an auxiliary excitatory neuron that makes F fire in isolation fires at ~50 Hz. The maximal AMPA conductance from the auxiliary excitatory neuron to F is $-g_e = 0.2\,mS/cm^2$ (see the next paragraph for a description of the AMPA synapses). Finally, US influences VIP activity by increasing its $I_{app}$ set to $5\,\mu A/cm^2$. To make *Appendix 1—figure 3*, we also considered a variation of the model with PV interneurons affected by US, instead of CS, as reported in *Krabbe et al., 2019*.

## Network connectivity and synaptic currents

We modeled the network connectivity as presented in *Figure 2B*, derived from the prominent functional, instead of structural, connections reported in *Krabbe et al., 2019*. We have a total of 9 types of projections between neurons: 6 inhibitory (VIP → PV, VIP → SOM, PV → F, PV → ECS, SOM → F, SOM → ECS), 3 excitatory (ECS → F, F → PV, F → VIP).

All inhibitory synapses are described as GABAa currents ($I_{GABAa}$) using a Hodgkin-Huxley-type conductance, as formulated in *Olufsen et al., 2003*:

$$I_{GABAa} = -g_i s_i \left(V - E_i\right).$$

The maximal GABAa conductance VIP → PV is $\bar{g}_i = 1/N_{VIP}\,mS/cm^2$, VIP → SOM is $\bar{g}_i = 1/N_{SOM}\,mS/cm^2$, PV → F is $\bar{g}_i = 0.5/N_{PV}\,mS/cm^2$, PV → ECS is $\bar{g}_i = 0.4/N_{PV}\,mS/cm^2$, SOM → F is $\bar{g}_i = 0.4/N_{SOM}\,mS/cm^2$, and SOM → ECS is $\bar{g}_i = 0.4/N_{SOM}\,mS/cm^2$, where $N_{VIP}$, $N_{PV}$, $N_{SOM}$ are the number of VIP, PV, and SOM cells, respectively, in the network. The GABAa current reversal potential ($E_i$) is set to $-80\,mV$, as common in the modeling literature (*Jensen et al., 2005*; *Traub et al., 2005*; *Chartove et al., 2020*). The variable $s_i$ represents the gating variable for inhibitory GABAa synaptic transmission, where $i$ stands for inhibitory synapse. The contribution of an inhibitory synapse to a specific postsynaptic inhibitory neuron $j$ in the network takes the following form:

$$s_i = \sum_k S_{i_k i_j}. \tag{4}$$

The contribution to a specific postsynaptic excitatory neuron $m$ in the network, reads as follows:

$$s_i = \sum_k S_{i_k e_m}, \tag{5}$$

where $k$ indexes the presynaptic inhibitory neurons. The variable $S_{i_k i_j}$ in *Equation 4* describes the kinetics of the gating variables from the inhibitory presynaptic neuron $k^{th}$ to the inhibitory postsynaptic neuron $j$. This variable evolves in time according to:

$$\frac{dS_{i_k i_j}}{dt} = g_{GABAa}\left(V_k\right)\left(1 - S_{i_k i_j}\right) - \frac{S_{i_k i_j}}{\tau_{i_k}}.$$

Similarly, the kinetics of the gating activation variable $S_{i_k e_m}$ in *Equation 5* from the $k^{th}$ interneuron to the postsynaptic excitatory neuron $m$ is formulated as:

$$\frac{dS_{i_k e_m}}{dt} = g_{GABAa}\left(V_k\right)\left(1 - S_{i_k e_m}\right) - \frac{S_{i_k e_m}}{\tau_{i_k}}.$$

The GABAa decay time constant ($\tau_{I_k}$) is a constant that depends on the type of presynaptic inter-neuron. The rate functions for the open state of the GABAa receptor ($g_{GABAa}\left(V_k\right)$) has a specific form based on the presynaptic cell type $k$. More specifically,

$$\text{For } k = VIP: \ g_{GABAa}\left(V_k\right) = 2\left(1 + \tanh\left(\frac{V_k}{4}\right)\right), \tau_{I_k} = 10\,ms$$

$$\text{For } k = PV: \ g_{GABAa}\left(V_k\right) = \frac{15}{2}\left(1 + \tanh\left(\frac{V_k}{0.1}\right)\right), \tau_{I_k} = 8.3\,ms$$

$$\text{For } k = SOM: \ g_{GABAa}\left(V_k\right) = \frac{5}{2}\left(1 + \tanh\left(\frac{V_k}{0.1}\right)\right), \tau_{I_k} = 20\,ms.$$

All excitatory synapses are described as AMPA currents ($I_{AMPA}$) using a Hodgkin-Huxley-type conductance, as formulated in *Olufsen et al., 2003*:

$$I_{AMPA} = -g_e s_e \left(V - E_e\right).$$

At the beginning of the fear conditioning paradigm there is no connection from ECS to F, that is the maximal AMPA conductance ECS → F is $\bar{g}_e = 0\,mS/cm^2$. Since ECS to F is a plastic connection (see paragraph related to synaptic plasticity), it evolves over time up to a maximum of $\bar{g}_e = 0.18\,mS/cm^2$. The maximal AMPA conductance F → PV is $\bar{g}_e = 0.5\,mS/cm^2$, and F → VIP is $\bar{g}_e = 0.01\,mS/cm^2$. In the case of the plastic F to VIP cell connections (see Appendix 1), the F → VIP conductances evolve over time up to a maximum of $\bar{g}_e = 0.04\,mS/cm^2$. The AMPA current reversal potential ($E_e$) is set to $0\,mV$.

The variable $s_e$ represents the gating variable for excitatory AMPA synaptic transmission, where $e$ stands for excitatory synapse. For a specific postsynaptic excitatory neuron $m$ in the network:

$$s_e = \sum_k S_{e_k e_m}.$$

For a specific postsynaptic inhibitory neuron $j$ in the network:

$$s_i = \sum_k S_{e_k i_j},$$

where $k$ indexes the presynaptic excitatory neurons.

The variable $S_{e_k e_m}$ describes the kinetics of the gating variables from the excitatory presynaptic neuron $k^{th}$ to the excitatory postsynaptic neuron $m$. This variable evolves in time according to:

$$\frac{dS_{e_k e_m}}{dt} = g_{AMPA}\left(V_k\right)\left(1 - S_{e_k e_m}\right) - \frac{S_{e_k e_m}}{\tau_e}.$$

Similarly, the kinetics of the synaptic activation variable of the $k^{th}$ excitatory neuron to the inhibitory neuron $j$ is denoted by $S_{e_k i_j}$ and is formulated as:

$$\frac{dS_{e_k i_j}}{dt} = g_{AMPA}\left(V_k\right)\left(1 - S_{e_k i_j}\right) - \frac{S_{e_k i_j}}{\tau_e}.$$

The time-constant of decay for the AMPA synapse is $\tau_e = 2\,ms$. The rate functions for the open state of the AMPA receptor ($g_{AMPA}\left(V_k\right)$) follows the mathematical formulation:

$$g_{AMPA}\left(V_k\right) = 5\left(1 + \tanh\left(\frac{V_k}{4}\right)\right) \ k = F\,ECS.$$

## Synaptic plasticity

Fear conditioning is a paradigm able to create associative learning between the neutral (CS) and the aversive (US) stimuli. Synaptic plasticity is thought to be at the basis of associative learning. In our work, synaptic plasticity takes the form of spike-timing-dependent plasticity (*Song et al., 2000*;

*Lee et al., 2009*), where synaptic modifications are enforced at the synapse from ECS to F at each presynaptic (ECS) and postsynaptic (F) neuron spikes. Synaptic modification is generated in the model through two auxiliary functions: P, used for potentiation, and M, used for depression, as in standard STDP models. The major difference between our use of STDP and some others is that, for each pre-synaptic spike, we take into account all the post-synaptic spikes with which it could potentially interact; similarly for each post-synaptic spike, we take into account all presynaptic spikes. P and M are initialized at zero and are updated at each presynaptic and postsynaptic neuron spike, respectively. Between spikes, they exponentially decay to zero (see *Appendix 1—figure 2* and *Appendix 1—figure 4*). The update is described as follows. At each postsynaptic neuron (F) spike, $M(t)$ is decremented by an amount $A_- = 0.005$, i.e., $M(t) = M(t) - A_-$. At each presynaptic neuron (ECS) spike, $P(t)$ is incremented by an amount $A_+ = 0.005$, i.e., $P(t) = P(t) + A_+$. This Hebbian plasticity rule is depression-dominant when $A_+ = A_-$ and $\tau_+ < \tau_-$, as formulated in this work. The exponential decay is described by the following equations:

$$\tau_- \frac{dM}{dt} = -M$$

$$\tau_+ \frac{dP}{dt} = -P,$$

with $\tau_- = 28\,ms$ and $\tau_+ = 14\,ms$. Every time the synapse receives an ECS action potential at time $t$, its maximal conductance is weakened according to $\bar{g} = -g + M(t)$. If $\bar{g} < 0$, $\bar{g}$ is set to zero. If F fires an action potential at time $t$, the synapse maximal conductance is strengthened according to $-g = \bar{g} + P(t)$. If this strengthening makes $\bar{g} > \bar{g}_{max}$, then $\bar{g}$ is set to $-g_{max} = 0.18\,mS/cm^2$ (see *Appendix 1—figure 2A*, right, for an example). See Appendix 1 for a visualization of M and P and consequences of the depression-dominated rule.

In Appendix 1, we introduce a plastic connection from the fear neuron F to the VIP interneurons. For that specific synapse, $A_+ = 0.00065$ and $A_- = 0.0003$, while the decay time constants are as above. In this case, the rule is not depression-dominant.

## Model simulations

Our network models were programmed in C++and compiled using g++compiler (Apple clang version 14.0.0) on macOS Monterey version 12.5.1. The differential equations were integrated using a fourth-order Runge Kutta algorithm. The integration time step was 0.05ms. Model output is graphed and analyzed using MATLAB, Version R2022a. Simulation codes are made freely available (see Resource Availability section).

## Local field potentials and spectral analysis

### Modeling LFP

One measure of neuronal population activity in the BLA is the LFP. We considered as an LFP proxy as the linear sum of all the AMPA, GABA, NaP-, D-, and H- currents in the network. The D-current is in the VIP interneurons, and NaP-current and H-current are in SOM interneurons.

### Spectral analysis

Stationarity of the network before and after fear conditioning is ensured after 2000ms. Thus, to ensure elimination of transients due to initial conditions, we discard the first 2000ms of LFP signals. LFP's power spectra are calculated using the Thomson's multitaper power spectral density estimate (MATLAB function pmtm; *Bokil et al., 2007*) for frequencies ranging from 0.1 to 70 Hz. Analysis codes are made freely available (see Resource Availability section).

## Acknowledgements

Research reported here was supported by the National Institute of Neurological Disorders and Stroke of the National Institute of Health under Award Number U01NS122082 to NK and DBA. We would like to thank Jamie Maguire and Leon Reijmers for their invaluable discussions during the development of this work, and Daniel Gibson for his insightful feedback on the manuscript.

## Additional information

### Funding

| Funder | Grant reference number | Author |
|---|---|---|
| National Institute of Neurological Disorders and Stroke | UO1NS122082 | Don B Arnold Nancy Kopell |

The funders had no role in study design, data collection and interpretation, or the decision to submit the work for publication.

### Author contributions

Anna Cattani, Conceptualization, Data curation, Software, Formal analysis, Investigation, Methodology, Writing – original draft, Writing – review and editing; Don B Arnold, Conceptualization, Funding acquisition, Visualization; Michelle McCarthy, Conceptualization, Software, Formal analysis, Supervision, Investigation, Visualization, Methodology, Writing – original draft, Writing – review and editing; Nancy Kopell, Conceptualization, Supervision, Funding acquisition, Investigation, Visualization, Methodology, Writing – original draft, Writing – review and editing

### Author ORCIDs

Anna Cattani ⬥ https://orcid.org/0000-0003-2317-1737
Don B Arnold ⬥ https://orcid.org/0000-0001-7378-1440

Reviewer #1 (Public review): https://doi.org/10.7554/eLife.89519.4.sa1
Author response https://doi.org/10.7554/eLife.89519.4.sa2

## Additional files

### Supplementary files

• MDAR checklist

### Data availability

The computer simulation code created for this paper, as well as the code to make the figures presented, are available online at https://github.com/annacatt/Basolateral_amygdala_oscillations_enable_fear_learning_in_a_biophysical_model (copy archived at *Cattani, 2024*). The simulated data that support the findings of this study are available at https://doi.org/10.5061/dryad.pvmcvdnr2.

The following dataset was generated:

| Author(s) | Year | Dataset title | Dataset URL | Database and Identifier |
|---|---|---|---|---|
| Cattani A | 2023 | Basolateral amygdala oscillations enable fear learning in a biophysical model | https://doi.org/10.5061/dryad.pvmcvdnr2 | Dryad Digital Repository, 10.5061/dryad.pvmcvdnr2 |

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

## Appendix 1

### VIP and SOM interneurons fire at gamma nested low theta and high theta, respectively

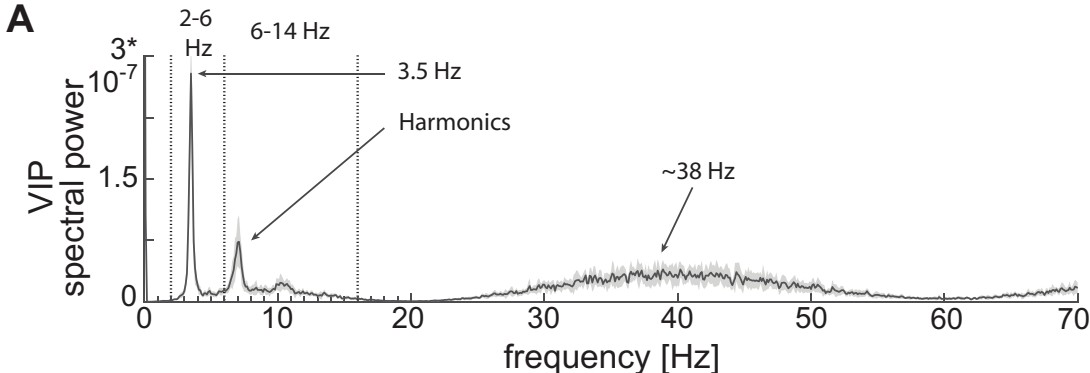

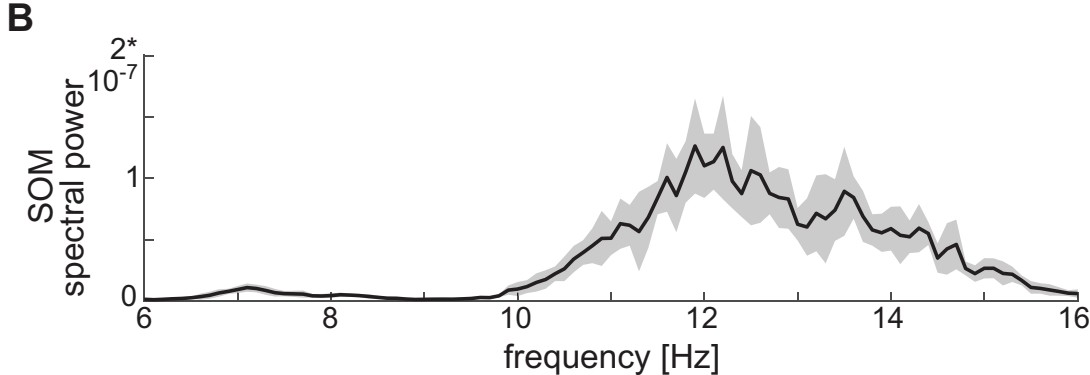

**Appendix 1—figure 1.** Power spectra of neuronal spiking activity at baseline. (**A**) Power spectrum of VIP cell showing a first peak at low theta (~3.5 Hz), plus harmonics, and a second peak at gamma (~38 Hz). (**B**) SOM cell power spectrum showing a peak at high theta (~12 Hz). Both panels show the mean (black curves) and standard deviation (black shaded areas) of the power spectra across 10 network realizations.

### ECS and F activity patterns determine overall potentiation or depression

The STDP model used here considers the whole history of ECS and F spikes by exploiting functions P, used for potentiation, and M, used for depression (*Song et al., 2000*). P evolution in time is determined by ECS (presynaptic neuron) spiking activity, while M is shaped by F (postsynaptic neuron) spiking activity. As detailed in the section 'Synaptic plasticity' in the Materials and methods, these two functions change at the time a neuron spikes and then relax exponentially towards the equilibrium (0) (see *Appendix 1—figure 2*).

When both ECS and F are firing at gamma (as shown in *Appendix 1—figure 2B*, right, in the first 100ms), both P and M build up, but since M has a longer relaxation time it builds up more than P, such that after ~100ms of gamma depression dominates. Note that M and P decay back to zero if there is a pause in spiking for a theta cycle (*Appendix 1—figure 2B*, right, in between ~8150 and 8250ms). The actual change of synaptic conductance is computed as follows. When F spikes, the instantaneous value of P determines the amount of potentiation the synapse from ECS to F undergoes. By contrast, when ECS spikes, the instantaneous value of M determines the amount of depression that weakens that synapse. *Appendix 1—figure 2* presents the P and M functions, along with the network dynamics and the evolution of the conductance from ECS to F, for the full network (*Figure 2*), the only-PV network (*Figure 4A*), and the PV/VIP network (*Figure 4B*) in the presence of CS and US.

By using a depression-dominated rule (*Appendix 1—figure 2A*), we show in the main text that the ECS to F conductance overall potentiates in presence of the full network after the US onset (*Figure 2*). *Appendix 1—figure 2B* shows that overall potentiation wins over depression because: (i) during the long disinhibition windows at low theta provided by VIP, both ECS and F are active and ECS fires most of the time slightly before F; despite M being slightly stronger than P, the pre-post fine timing ensures potentiation; (ii) M relaxes towards the equilibrium during the silent ECS-F phase of the low theta rhythm, and thus the potentiation acquired during the active ECS and F phase builds up over time.

In the PV-only network (*Appendix 1—figure 2C*), with PV in a low excitation regime such that it can entrain with F in a coordinated gamma rhythm (PING), we show in the main text that there is overall depression in the conductance from ECS to F (*Figure 4A*) because PING is not periodically interrupted. Indeed, PING without interruptions makes the M function saturate. By contrast, the P function shows only a few jumps and long periods at the equilibrium due to a few ECS spikes. Thus, a weak potentiation happens at each F spike, but it is counterbalanced by the strong depression at each (despite few) ECS spike, leading to overall depression.

In the network with PV and VIP (*Appendix 1—figure 2D*), we find that VIP cell reduces the excitation of the PV cell (which is set at high excitation, unless otherwise specified) during its active low theta phase, enabling the PV cell to participate in PING. Secondly, it provides periodic interruptions in PING (*Figure 4B*). *Appendix 1—figure 2D* shows that potentiation in the conductance from ECS to F arises during the active VIP phases because of the ECS-F fine timing. However, during the silent VIP phases, ECS is still active due to the weaker inhibition from PV and the lack of inhibition from SOM. At each ECS spike, a non-negligible depression happens despite the M function relaxing. In conclusion, the depression brought about by ECS spikes during the silent VIP phases counteracts the potentiation acquired during the active VIP phases such that, overall, no significant potentiation takes place.

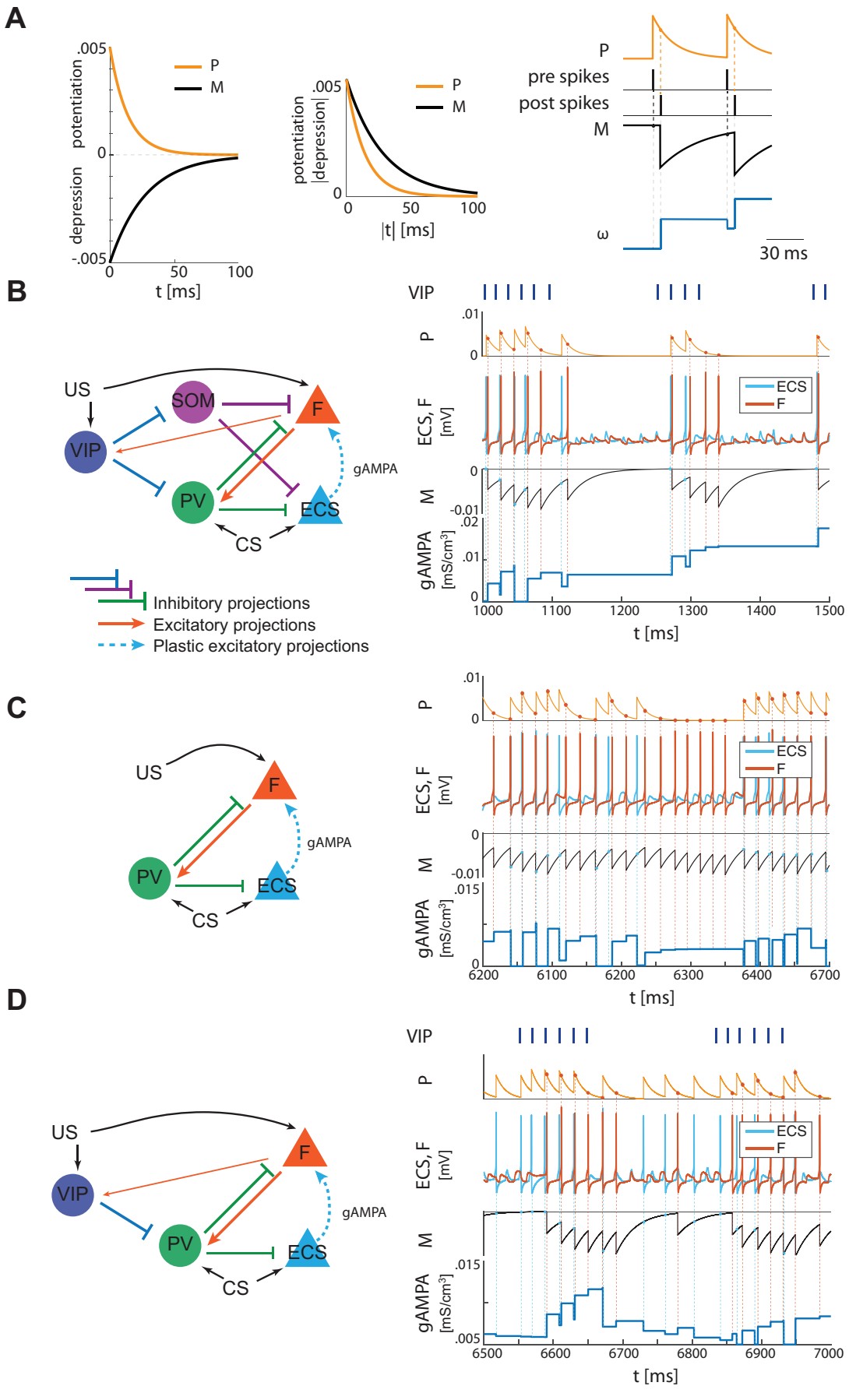

**Appendix 1—figure 2.** Plasticity rule and detailed representation of how plasticity works in the three network configurations shown in *Figure 2B* and *Figure 4A and B*. (**A**) Left, STDP potentiation (**P**) and depression (**M**) functions. The area underlying depression is larger than the one of potentiation, thus providing a depression-dominated rule. The potentiation and depression curves are used to compute the instantaneous change of the synaptic conductance as a function of the spike time difference between pre and postsynaptic neurons, as detailed in the section 'Synaptic plasticity' in Materials and methods, and in Appendix 1. Right, a representative spike pattern of pre and postsynaptic neurons alongside the resulting P and M functions and the evolution of the pre-post synaptic conductance. (**B**) Left, full network. Right, ECS (pre) and F (post) spiking activities over 500 ms (extracted from *Figure 2*) with their respective M and P functions, which determine how the AMPA conductance from ECS to F evolves in time. (**C**) Left, only-PV network at a low excitation level. Right, ECS and F dynamics as in 500ms extracted from *Figure 4A* with P, M, and AMPA conductance unfolded over time. (**D**) Left, network with VIP and PV. Right, ECS and F dynamics as in 500ms extracted from *Figure 4B*, alongside P, M, and AMPA conductance over time.

## An alternative network configuration characterized by US input to PV, instead of CS, also learns the association between CS and fear

We constrained the BLA network in *Figure 2* with CS input to the PV interneuron, as reported in *Krabbe et al., 2018*. However, (*Krabbe et al., 2019*) notes that a class of PV interneurons may be responding to US rather than CS. *Appendix 1—figure 3* presents the results obtained with this variation in the model (see *Figure 3A and B* for comparison) and shows that all the network realizations learn the association between CS and fear. In the model, the PING rhythm between PV and F is the crucial component for establishing fine timing between ECS and F, which is necessary for learning. Having PV responding to the same input as F, that is US, facilitates their entrainment in PING and, thus, successful fear learning.

We model the VIP interneuron as affected by US; in addition, (*Krabbe et al., 2019*) reports that a substantial proportion of them is mildly activated by CS. Replacing the US by CS does not change the input to VIP cells, which is modeled by the same constant applied current. Thus, the VIP CS-induced activity is a bursting activity at low theta, similar to the one elicited by US in *Figure 2*.

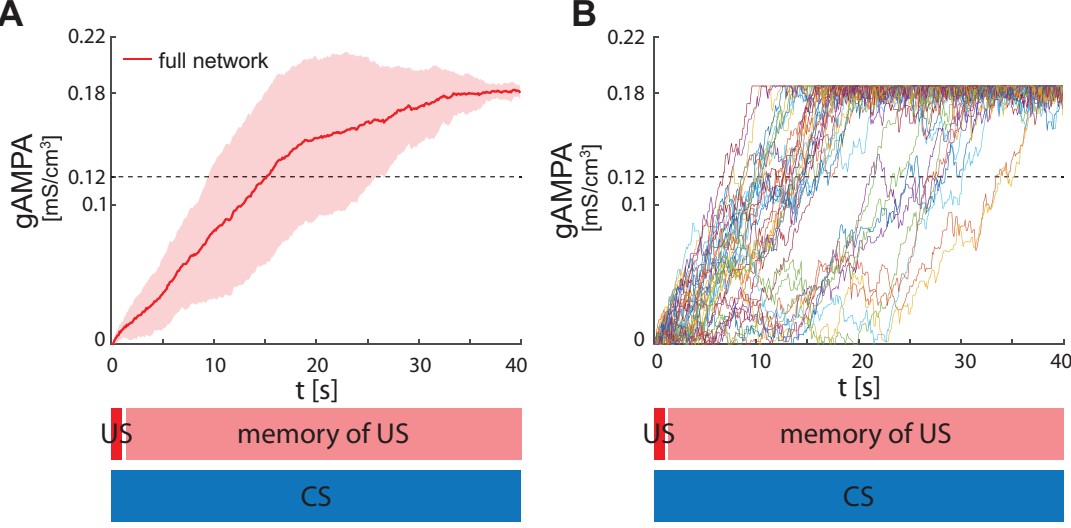

**Appendix 1—figure 3.** ECS to F conductance across network realizations in BLA characterized by US, instead of CS, input to PV interneuron. (**A**) Mean (color-coded curves) and standard deviation (color-coded shaded areas) of the AMPA conductance (gAMPA) from ECS to F across 40 network realizations over 40 seconds. (**B**) Evolution in time of the AMPA conductance for the 40 full network realizations in A.

## Classical Hebbian plasticity rule, unlike the depression-dominated one, shows potentiation even with no strict pre and postsynaptic spike timing

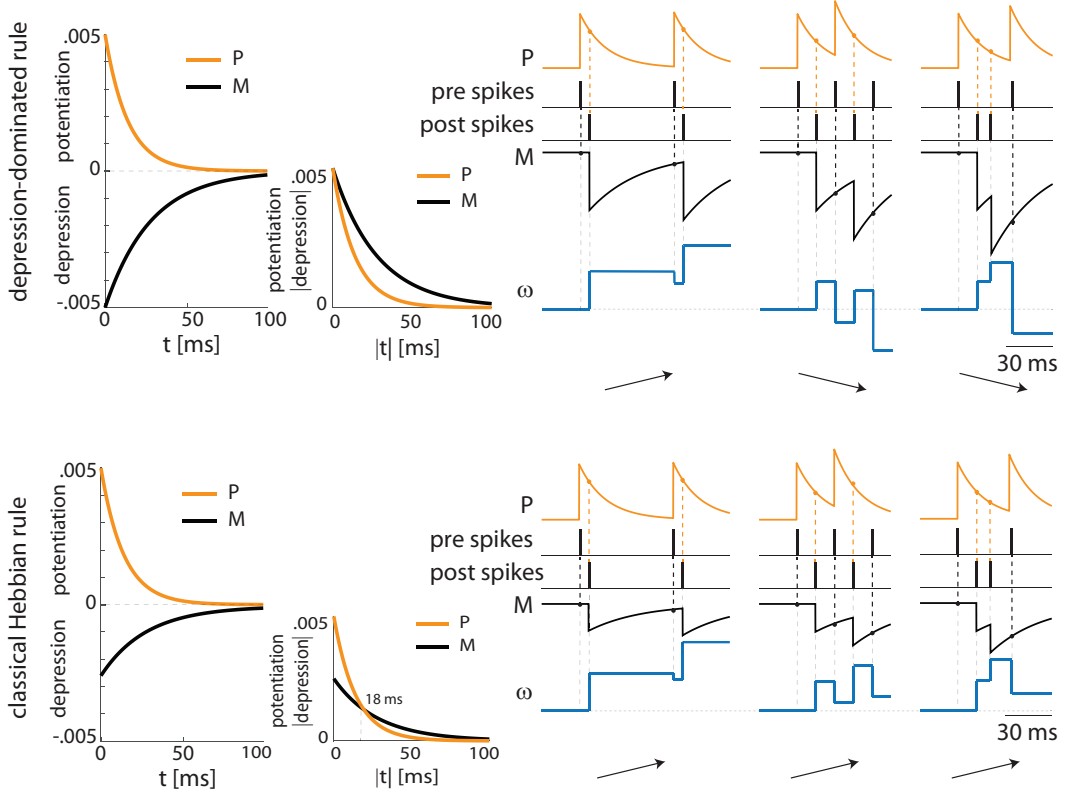

**Appendix 1—figure 4.** Depression-dominated and classical Hebbian plasticity rules may provide opposite potentiation/depression profiles. Top, depression-dominated rule (as in *Appendix 1—figure 2A*) with three examples of pre and postsynaptic spike patterns. Only the first one, which shows a consistent pre-post timing, shows overall potentiation. The remaining two spike patterns lead to depression. Bottom, classical Hebbian rule characterized by a smaller maximal amplitude for depression than potentiation. In agreement with the depression-dominated rule, the classical rule shows potentiation in the case of the pre and postsynaptic neurons showing correct pre-post spike timing. However, the classical rule shows potentiation also in the remaining two examples where there is no-consistent pre-post timing, given that the pre and postsynaptic neurons fire at a frequency higher than 55 Hz.

## A higher low theta power increase emerges in LFP approximated with the sum of the absolute values of the currents compared to their linear sum

Given that our BLA network comprises a few neurons described as single-compartment cells with no spatial extension and location, the LFP cannot be computed directly from our model's read-outs. In the main text, we choose as an LFP proxy the linear sum of the AMPA, GABA, and P-/H-/D-currents. We note that if the LFP is modeled as the sum of the absolute value of the currents, as suggested by *Mazzoni et al., 2008*, *Mazzoni et al., 2015*, an even higher low theta power increase arises after fear conditioning compared to the linear sum. Differences in the power spectra also arise if other LFP proxies (e.g. only AMPA currents, only GABA currents) are considered. A principled description of an LFP proxy would require modeling the three-dimensional BLA anatomy, including that of the interneurons VIP and SOM; this is outside the scope of the current paper. (See *Feng et al., 2019* for a related project in the BLA).

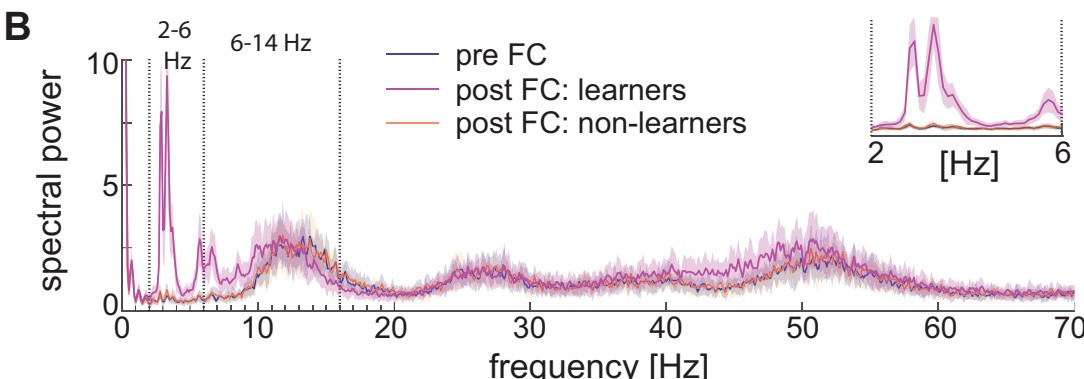

**Appendix 1—figure 5.** Spectral properties pre versus post fear conditioning of LFP approximated with the sum of the absolute values of AMPA, GABA, D-, NaP-, and H-currents. Power spectra before fear conditioning (blue) and after successful (purple) and non-successful fear conditioning (orange); top, right: inset between 2 and 6 Hz. Blue and orange curves closely overlap.

