## [Editor Report · eLife Assessment]

This **valuable** modeling study explores how biophysical properties of different interneuron subtypes in the basolateral amygdala (BLA) enable production of oscillations that facilitate functions such as spike-timing-dependent plasticity. Simulated networks provide **solid** evidence that highlights the importance of interactions between interneurons for some forms of spike-timing dependent plasticity. This work will likely be of interest to investigators studying interactions among interneurons, rhythms in the amygdala, and mechanisms of plasticity thought to underlie associative learning.

---

## [Referee Report · Reviewer #1 (Public review)]

Plasticity in the basolateral amygdala (BLA) is thought to underlie the formation of associative memories between neutral and aversive stimuli, i.e. fear memory. Concomitantly, fear learning modifies the expression of BLA theta rhythms, which may be supported by local interneurons. Several of these interneuron subtypes, PV+, SOM+, and VIP+, have been implicated in the acquisition of fear memory. However, it was unclear how they might act synergistically to produce BLA rhythms that structure the spiking of principal neurons so as to promote plasticity. Cattani et al. explored this question using small network models of biophysically detailed interneurons and principal neurons.

Using this approach, the authors had four principal findings:

(1) Intrinsic conductances in VIP+ interneurons generate a slow theta rhythm that periodically inhibits PV+ and SOM+ interneurons, while disinhibiting principal neurons.

(2) A gamma rhythm arising from the interaction between PV+ and principal neurons establishes the precise timing needed for spike-timing-dependent plasticity.

(3) Removal of any of the interneuron subtypes abolishes conditioning-related plasticity.

(4) Learning-related changes in principal cell connectivity enhance expression of slow theta in the local field potential.

The strength of this work is that it explores the role of multiple interneuron subtypes in the formation of associative plasticity in the basolateral amygdala. The authors use biophysically detailed cell models that capture many of their core electrophysiological features, which helps translate their results into concrete hypotheses that can be tested in vivo. Moreover, they try to align the connectivity and afferent drive of their model with those found experimentally.

A drawback to this study is the construction of the afferent drive to the network, which does not elicit activities that are consistent with the majority of those observed to similar stimuli. The authors discuss this issue in depth, and provide potential mechanisms that may overcome it.

Setting aside the issues with the conditioning protocol, the study offers a model for the generation of multiple rhythms in the BLA that is ripe for experimental testing. The most promising avenue would be in vivo experiments testing the role of local VIP+ neurons in the generation of slow theta. That would go a long way to resolving whether BLA theta is locally generated or inherited from medial prefrontal cortex or ventral hippocampus afferents.

The broader importance of this work is that it illustrates that we must examine the function of neurons not just in terms of their behavioral correlates, but by their effects on the microcircuit they are embedded within. No one cell type is instrumental in producing fear learning in the BLA. Each contributes to the orchestration of network activity to produce plasticity. Moreover, this study reinforces a growing literature highlighting the crucial role of theta and gamma rhythms in BLA function.

---

## [Author Response]

The following is the authors’ response to the current reviews.

We thank the Reviewer for all their effort and suggestions over multiple drafts. Their comments have encouraged us to read and think more deeply about the issue under discussion (BLA spiking in response to CS/US inputs), and to find the papers whose contents we think provide a potential solution. We agree that there is more to understand about the mechanisms underlying associative learning in the BLA. We offer our paper as providing a new way of understanding the role of circuit dynamics (rhythms) in guiding associative learning via STDP. As we pointed out in our response to the previous review, the issue highlighted by the Reviewer is an issue for the entire field of associative learning in BLA: our discussion of the issue suggests why the experimentally observed BLA spiking in response to CS inputs, performed in the absence of US inputs (as done in the papers cited by the Reviewer), may not be what occurs in the presence of the US. Since our explanation involves the role of neuromodulators, such as ACh and dopamine, the suggestion is open to further testing.

The following is the authors’ response to the original reviews.

**Reviewer #1:**
Public Review’s only objection: “Deficient in this study is the construction of the afferent drive to the network, which does elicit activities that are consistent with those observed to similar stimuli. It still remains to be demonstrated that their mechanism promotes plasticity for training protocols that emulate the kinds of activities observed in the BLA during fear conditioning.”

Recommendations for the Authors: “The authors have successfully addressed most of my concerns. I commend them for their thorough response. The one nagging issue is the unrealistic activation used to drive CS and US activation in their network. While I agree that their stimulus parameters are consistent with a contextual fear task, or one that uses an olfactory CS, this was not the focus of their study as originally conceived. Moreover, the types of activation observed in response to auditory cues, which is the focus of their study, do not follow what is reported experimentally. Thus, I stand by the critique that the proposed mechanism has not been demonstrated to work for the conditioning task which the authors sought to emulate (Krabbe et al. 2019). Frustratingly, addressing this is simple: run the model with ECS neurons driven so that they fire bursts of action potentials every ~1 sec for 30 sec, and with the US activation noncontiguous with that. If the model does not produce plasticity in this case, then it suggests that the mechanisms embedded in the model are not sufficient, and more work is needed to identify them. While 'memory' effects are possible that could extend the temporal contiguity of the CS and US, the authors need to provide experimental evidence for this occurring in the BLA under similar conditions if they want to invoke it in their model.

(1) Fair response. I accept the authors arguments and changes.(2) The authors rightly point out that the simulated afferents need not perfectly match the time courses of the peripheral inputs, since what the amygdala receives them indirectly via the thalamus, cortex, etc. However, it is known how amygdala neurons respond to such stimuli, so it behooves the authors to incorporate that fact into their model.Quirk et al. 1997 show that the response to the tone plummets after the first 100 ms in Figs 5A and 6B. The Herry et al. 2007 paper emphasizes the transient response to tone pips, with spiking falling back to a poisson low firing rate baseline outside of the time when the pip is delivered.Regarding potential metabotropic glutamate activation, the stimulus in Whittington et al. 1995 was electrical stimulation at 100 Hz that would synchronously activate a large volume of tissue, which is far outside the physiological norm. I appreciate that metabotropic glutamate receptors may play a role here, but ultimately the model depends upon spiking activity for the plastic process to occur, and to the best of my knowledge the spiking activity in BLA in response to a sustained, unconditioned tone, is brief (see also Quirk, Repa, and Ledoux 1995). Perhaps a better justification for the authors would be Bordi and Ledoux 1992, which found that 18% of auditory responsive neurons showed a 'sustained' response, but the sustained response neurons appear to show much weaker responses than those with transient ones (Fig 2). I am willing to say that their paper IS relevant to contextual fear, but that is not what the authors set out to do.(3) Fair response.(4) Very good response!Minor points: All points were addressed.”

We thank Reviewer 1 (R1) for the positive feedback and also for pointing out that, in R1’s opinion, there is still a nagging issue related to the activation in response to CS we modeled. In (Krabbe et al., 2019), CS is a pulsed input and US is delivered right after the CS offset. The current objection of R1 is that instead, we are modeling CS and US as continuous and overlapping. R1 suggested that we add the actual input and see if they will produce the desired outputs. The answer is simple: it will not work because we need the effects of CS and US on pyramidal cells to overlap. We note that the fear learning community appears to agree with us that such contingency is necessary for synaptic plasticity (Sun et al., 2020; Palchaudhuri et al., 2024). To the best of our understanding, the source of that overlap is not understood in the community, and the gap has been much noticed (Sun et al., 2020). We do note, however, that STDP may not be the only kind of plasticity in fear learning (Li et al., 2009; Kim et al., 2013, 2016).

It is important to emphasize that it is not the aim of our paper to model the origin of the overlap. Rather, our intent is to demonstrate the roles of brain rhythms in producing the appropriate timing for STDP, assuming that ECS and F cells can continue to be active after the offset of CS and US, respectively. This assumption is very close to how the field now treats the plasticity, even for auditory fear conditioning (Sun et al., 2020). Thus, our methodology does not contradict known results. However, the question raised by R1 is indeed very interesting, if not the point of our paper. Hence, below we give details about why our hypothesis is reasonable.

Several papers (Quirk, Repa and LeDoux, 1995; Herry et al, 2007; Bordi and Ledoux 1992) show that the pips in auditory fear conditioning increase the activity of some BLA neurons: after an initial transient, the overall spike rate is still higher than baseline activity. As R1 points out, we did not model the transient increase in BLA spiking activity that occurs in response to each pip in the auditory fear conditioning paradigm. However, we did model the low-level sustained activity that occurs in between pips of the CS in the absence of US (Quirk, Repa and LeDoux, 1995, Fig. 2) and after CS offset (see Fig. 2B, left hand part of our manuscript). We read the data of Quirk et al., 1995 as suggesting that the low-level activity can be sustained for some indefinite time after a pip (cut off of recording was at 500 ms with no noticeable decrease in activity). As such, even if the pips and the US do not overlap in time, as in (Krabbe et al., 2019), the spiking of the ECS can be sustained after CS offset and thus overlap with US, a condition necessary in our model for plasticity through STDP. In Herry et al., 2007 Fig. 3 shows that BLA neurons respond to a pip at the population level with a transient increase in spiking and return to a baseline Poisson firing rate. However, a subset of cells continues to fire at an increased-over-baseline rate after the transient effect wears off (Fig. 3C, top few neurons) and this increased rate extends to the end of the recording time (here ~ 300 ms). These are the cells we consider to be ECS in our model. In Quirk et al., 1997, Fig. 5A also shows sustained low level activity of neurons in BLA in response to a pip. The low-level activity is shown to increase after fear learning, as is also the case in our model since ECS now entrains F so that there are more pyramidal cells spiking in response to CS. The question remains as to whether the spiking is sustained long enough and at a high enough rate for STDP to take place when US is presented sometime after the stop of the CS.

Experimental recordings cannot speak to the rate of spiking of BLA neurons during US due to recording interference from the shock. However, evidence seems to suggest that ECS activity should increase during the US due to the release of acetylcholine (ACh) from neurons in the basal forebrain (BF) (Rajebhosale et al., 2024). Pyramidal cells of the BLA robustly express M1 muscarinic ACh receptors (Muller et al., 2013; McDonald and Mott, 2021). Thus, ACh from BF should elicit a depolarization in pyramidal cells. Indeed, the pairing of ACh with even low levels of spiking of BLA neurons results in a membrane depolarization that can last 7 – 10 s (Unal et al., 2015). This should induce higher spiking rates and more sustained activity in the ECS and F neurons during and after the presentation of US, thus ensuring a concomitant activation of ECS and fear (F) neurons necessary for STDP to take place. Other modulators, including dopamine, may also play a role in producing the sustained activity. Activation of US leads to increased dopamine release in the BLA (Harmer and Phillips, 1999; Suzuki et al., 2002). D1 receptors are known to increase the membrane excitability of BLA projection neurons by lowering their spiking threshold (Kröner et al., 2005). Thus, the activation of the US can lead to continued and higher firing rates of ECS and F. The effect of dopamine can last up to 20 minutes (Kröner et al., 2005). For CS-positive neurons, the ACh modulation coming from the firing of US may lead to a temporary extension of firing that is then amplified and continued by dopaminergic effects.

Hence, we suggest that a solution to the problem raised by R1 may be solved by considering the roles of ACh and dopamine in the BLA. The involvement of neuromodulators is consistent with the suggestion of (Sun et al., 2020). The model we have may be considered a “minimal” model that puts in by hand the overlap in activity due to the neuromodulation without explicitly modeling it. As R1 says, it is important for us to give the motivation of our hypotheses. We have used the simplest way to model overlap without assumptions about timing specificity in the overlap.

To account for these points in the manuscript, we first specified that we consider the effects of the US and CS inputs on the neuronal network as overlapping, while the actual inputs may not overlap. To do that, we added the following text:

(1) In the introduction:

“In this paper, we aim to show (1) How a variety of BLA interneurons (PV, SOM and VIP) lead to the creation of these rhythms and (2) How the interaction of the interneurons and the rhythms leads to the appropriate timing of the cells responding to the US and those responding to the CS to promote fear association through spike-timing-dependent plasticity (STDP). Since STDP requires overlap of the effects of the CS and US, and some conditioning paradigms do not have overlapping US and CS, we include as a hypothesis that the effects of the CS and US overlap even if the CS and US stimuli do not. In the Discussion, we suggest how neuromodulation by ACh and/or dopamine can provide such overlap. We create a biophysically detailed model of the BLA circuit involving all three types of interneurons and show how each may participate in producing the experimentally observed rhythms and interacting to produce the necessary timing for the fear learning.”

(2) In the Result section “With the depression-dominated plasticity rule, all interneuron types are needed to provide potentiation during fear learning”:

“The 40-second interval we consider has both ECS and F, as well as VIP and PV interneurons, active during the entire period: an initial bout of US is known to produce a long-lasting fear response beyond the offset of the US (Hole and Lorens, 1975) and to induce the release of neuromodulators. The latter, in particular acetylcholine and dopamine that are known to be released upon US presentation (Harmer and Phillips, 1999; Suzuki et al., 2002; Rajebhosale et al., 2024), may induce more sustained activity in the ECS, F, VIP, and PV neurons during and after the presentation of US, thus ensuring a concomitant activation of those neurons necessary for STDP to take place (see “Assumptions and predictions of the model” in the Discussion).”

(3) In the Discussion section “Synaptic plasticity in our model”:

“Synaptic plasticity is the mechanism underlying the association between neurons that respond to the neutral stimulus CS (ECS) and those that respond to fear (F), which instantiates the acquisition and expression of fear behavior. One form of experimentally observed long-term synaptic plasticity is spike-timing-dependent plasticity (STDP), which defines the amount of potentiation and depression for each pair of pre- and postsynaptic neuron spikes as a function of their relative timing (Bi and Poo, 2001; Caporale and Dan, 2008). All forms of STDP require that there be an overlap in the firing of the pre- and postsynaptic cells. In some fear learning paradigms, the US and the CS do not overlap. We address this below under “Assumptions and predictions of the model”, showing how the effects of US and CS on the spiking of the relevant neurons can overlap even in the absence of overlap of US and CS.”

To fully present our reasoning about the origin of the overlap of the effects of US and CS, we modified and added to the last paragraph of the Discussion section “Assumptions and predictions of the model”, which now reads as follows:

“Finally, our model requires the effect of the CS and US inputs on the BLA neuron activity to overlap in time in order to instantiate fear learning through STDP. Such a hypothesis, that learning uses spike-timing-dependent plasticity, is common in the modeling literature (Bi and Poo, 2001; Caporale and Dan, 2008; Markram et al., 2011). Current paradigms of fear conditioning include examples in which the CS and US stimuli do not overlap (Krabbe et al., 2019). Such a condition might seem to rule out the mechanisms in our paper. Nevertheless, the argument below suggests that the effects of the CS and US can cause an overlap in neuronal spiking of ECS, F, VIP, and SOM, even when CS and US inputs do not overlap.

Experimental recordings cannot speak to the rate of spiking of BLA neurons during US due to recording interference from the shock. However, evidence suggests that ECS activity should increase during the US due to the release of acetylcholine (ACh) from neurons in the basal forebrain (BF) (Rajebhosale et al., 2024). Pyramidal cells of the BLA robustly express M1 muscarinic ACh receptors (McDonald and Mott, 2021). Thus, ACh from BF should elicit a depolarization in pyramidal cells. Indeed, the pairing of ACh with even low levels of spiking of BLA neurons results in a membrane depolarization that can last 7 – 10 s (Unal et al., 2015). Other modulators, including dopamine, may also play a role in producing the sustained activity. Activation of US leads to increased dopamine release in the BLA (Harmer and Phillips, 1999; Suzuki et al., 2002). D1 receptors are known to increase the membrane excitability of BLA projection neurons by lowering their spiking threshold (Kröner et al., 2005). Thus, neuromodulator release should induce higher spiking rates and more sustained activity in the ECS and F neurons during and after the presentation of US, thus ensuring a concomitant activation of ECS and fear (F) neurons necessary for STDP to take place. Thus, the activation of the US can lead to continued and higher firing rates of ECS and F. The effect of dopamine can last up to 20 minutes (Kröner et al., 2005). For CS-positive neurons, the ACh modulation coming from the firing of US may lead to a temporary extension of firing that is then amplified and continued by dopaminergic effects.

Hence, we suggest that a solution to the problem apparently posed by the non-overlap US and CS in some paradigms of auditory fear conditioning (Krabbe et al., 2019) may be solved by considering the roles of ACh and dopamine in the BLA. The model we have may be considered a “minimal” model that puts in by hand the overlap in activity due to the neuromodulation without explicitly modeling it. We have used the simplest way to model overlap without assumptions about timing specificity in the overlap. We note that, even though ECS and F neurons have the ability to fire continuously when ACh and dopamine are involved, the participation of the interneurons enforces periodic silence needed for the depression-dominated STDP.”

In the Discussion (in section “Involvement of other brain structures”), we also acknowledged that the overlap between the effects of US and CS in the BLA may be provided by other brain structures by writing the following:

“In our model, the excitatory projection neurons and VIP and PV interneurons show sustained activity during and after the US presentation, thus allowing potentiation through STDP to take place. The medial prefrontal cortex and/or the hippocampus may provide the substrates for the continued firing of the BLA neurons after the 2-second US stimulation. We also discuss below that this network sustained activity may originate from neuromodulator release induced by US (see section “Assumptions and predictions of the model” in the Discussion).”

We also improved our discussion about the (Grewe et al., 2017) paper, which questions Hebbian plasticity in the context of fear conditioning based on several critiques. We included a new section in the Discussion entitled “Is STDP needed in fear conditioning?” to discuss those critiques and how our model may address them, which reads as follows:

“Is STDP needed in fear conditioning? The study in Grewe et al., 2017 questions the validity of the Hebbian model in establishing associative learning during fear conditioning. There are several critiques we discuss here. The first critique is that Hebbian plasticity does not explain the experimental finding showing that both upregulation and downregulation of stimulus-evoked responses are present between coactive neurons. The upregulation is provided by our model, so the issue is the downregulation, which is not addressed by our model. However, our model highlights that coactivity alone does not create potentiation; the fine timing of the pre- and postsynaptic spikes determines whether there is potentiation or depression. Here, we find that PING networks are instrumental in setting up the fine timing for potentiation. We suggest that networks not connected to produce the PING may undergo depression when coactive.

The second critique raised by Grewe et al., 2017 is that Hebbian plasticity alone does not explain why most of the cells exhibiting enhanced responses to the CS did not react to the US before fear conditioning. They suggest that neuromodulators may provide a third condition (besides the activity of the pre- and postsynaptic neurons) that changes the plasticity rule. Our model also does not explicitly address this experimental finding since it requires F to be initially activated by US in order for the fear association to be established. We agree that the fear cells described in Grewe et al. 2017 may be depolarized by the US without reaching the spiking threshold; however, with neuromodulation provided during the fear training, the same input can lead to spiking, enabling the conditions for Hebbian plasticity. Our discussions above about how neuromodulators affect excitability are relevant to this point. We do not exclude that other forms of plasticity may play a role during fear conditioning in cells not initially activated by the US, but this is not the topic of our modeling study.

The third critique raised by (Grewe et al., 2017) is that Hebbian plasticity cannot explain why the majority of cells that were US- and CS-responsive before training have a reduced CS-evoked response afterward. The reduced response happens over multiple exposures of CS without US; this can involve processes similar to those present in fear extinction, which require plasticity in further networks, especially involving the infralimbic cortex (Milad and Quirk, 2002; Burgos-Robles et al., 2007). An extension of our model could investigate such mechanisms. In the fourth critique, (Grewe et al., 2017) suggests that the Hebbian plasticity rule cannot easily account for the reduction of the responses of many CS+-responsive cells, but not of the CS−-responsive cells. We suggest that the circuits involving paradigms similar to fear extinction do not involve the CS- cells.

Overall, we agree with (Grewe et al., 2017) that neuromodulators play a crucial role in fear conditioning, especially in prolonging the US- and CS-encoding activity as discussed in (see section “Assumptions and predictions of the model” in the Discussion), or even participating in changing the details of the plasticity rule. A possible follow-up of our work involves investigating how fear ensembles form and modify through fear conditioning and later stages. This follow-up work may involve using a tri-conditional rule, as suggested in (Grewe et al., 2017), in which the potential role of neuromodulators is taken into account in the plasticity rule in addition to the pre- and postsynaptic neuron activity. Another direction is to investigate a possible relationship between neuromodulation and a depression-dominated Hebbian rule.”

Finally, we made additional minor changes to the manuscript:

(1) In the Result section “Interneurons interact to modulate fear neuron output”, we specified the following:

“The US input on the pyramidal cell and VIP interneuron is modeled as a Poisson spike train at ~ 50 Hz and an applied current, respectively. In the rest of the paper, we will use the words “US” as shorthand for “the effects of US”.”

(2) In the Result section “Interneuron rhythms provide the fine timing needed for depression dominated STDP to make the association between CS and fear”, we also reported the following:

“Similarly to the US, in the rest of the paper, we will use the words “CS” as shorthand for “the effects of CS”. In our simulations, CS is modeled as a Poisson spike train at ~ 50 Hz, independent of the US input. Thus, we hypothesize that the time structure of the inputs sometimes used for the training (e.g., a series of auditory pips) is not central to the formation of the plasticity in the network.”

**Reviewer #2 (Public Reviews):**
The authors of this study have investigated how oscillations may promote fear learning using a network model. They distinguished three types of rhythmic activities and implemented an STDP rule to the network aiming to understand the mechanisms underlying fear learning in the BLA.After the revision, the fundamental question, namely, whether the BLA networks can or cannot intrinsically generate any theta rhythms, is still unanswered. The author added this sentence to the revised version: "A recent experimental paper, (Antonoudiou et al., 2022), suggests that the BLA can intrinsically generate theta oscillations (3-12 Hz) detectable by LFP recordings under certain conditions, such as reduced inhibitory tone." In the cited paper, the authors studied gamma oscillations, and when they applied 10 uM Gabazine to the BLA slices observed rhythmic oscillations at theta frequencies. 10 uM Gabazine does not reduce the GABA-A receptor-mediated inhibition but eliminates it, resulting in rhythmic populations burst driven solely by excitatory cells. Thus, the results by Antonoudiou et al., 2022 contrast with, and do not support, the present study, which claims that rhythmic oscillations in the BLA depend on the function of interneurons. Thus, there is still no convincing evidence that BLA circuits can intrinsically generate theta oscillations in intact brain or acute slices. If one extrapolates from the hippocampal studies, then this is not surprising, as the hippocampal theta depends on extrahippocampal inputs, including, but not limited to the entorhinal afferents and medial septal projections (see Buzsaki, 2002). Similarly, respiratory related 4 Hz oscillations are also driven by extrinsic inputs. Therefore, at present, it is unclear which kind of physiologically relevant theta rhythm in the BLA networks has been modelled.

In our public reply to the Reviewer’s point, we reported the following:

(1) We kindly disagree that (Antonoudiou et al., 2022) contrasts with our study. (Antonoudiou et al., 2022) is a slice study showing that the BLA theta power (3-12 Hz) increases with gabazine compared to baseline. With all GABAergic currents omitted due to gabazine, the LFP is composed of excitatory currents and intrinsic currents. In our model, the high theta (6-12 Hz) comes from the spiking activity of the SOM cells, which increase their activity if the inhibition from VIP cells is removed. Thus, the model produces high theta in the presence of gabazine (see Fig. 1 in our replies to the Reviewers’ public comments). The model also shows that a PING rhythm is produced without gabazine, and that this rhythm goes away with gabazine because PING requires feedback inhibition from PV to fear cells. Thus, the high theta increase and gamma reduction with gabazine in the (Antonoudiou et al., 2022) paper can be reproduced in our model.

(2) We agree that (Antonoudiou et al., 2022) alone is not sufficient evidence that the BLA can produce low theta (3-6 Hz); we discussed a new paper (Bratsch-Prince et al., 2024) that provides further evidence of BLA ability to produce low theta and under what circumstances. The authors reported that intrinsic BLA theta is produced in slices with ACh stimulation (without needing external glutamate input) which, in vivo, would be provided by the basal forebrain (Rajebhosale et al., eLife, 2024) in response to salient stimuli. The low theta depends on muscarinic activation of CCK interneurons, a group of interneurons that overlaps with the VIP neurons in our model (Krabbe 2017; Mascagni and McDonald, 2003). We suspect that the low theta produced in (Bratsch-Prince et al., 2024) is the same as the low theta in our model. In future work, we will aim to show that ACh activates the BLA VIP cells, which are essential to the low theta generation in the network.

In the manuscript, we added to and modified the Discussion section “Where the rhythms originate, and by what mechanisms”. This text aims to better discuss (Antonoudiou et al. 2022) and introduce (Bratsch-Prince et al., 2024) with its connection to our hypothesis that the theta oscillations can be produced within the BLA. The new version is:

“Where the rhythms originate, and by what mechanisms. A recent experimental paper (Antonoudiou et al., 2022) suggests that the BLA can intrinsically generate theta oscillations (312 Hz) detectable by LFP recordings when inhibition is totally removed due to gabazine application. They draw this conclusion in mice by removing the hippocampus, which can volume conduct to BLA, and noticing that other nearby brain structures did not display any oscillatory activity. In our model, we note that when inhibition is removed, both AMPA and intrinsic currents contribute to the network dynamics and the LFP. Thus, interneurons with their specific intrinsic currents (i.e., D-current in the VIP interneurons, and NaP- and H- currents in SOM interneurons) can indeed affect the model LFP and support the generation of theta and gamma rhythms (Fig. 6G).

Another slice study, (Bratsch-Prince et al., 2024), shows that BLA is intrinsically capable of producing a low theta rhythm with ACh stimulation and without needing external glutamate input. ACh is produced in vivo by the basal forebrain in response to US (Rajebhosale et al., 2024). Although we did not explicitly include the BF and ACh modulation of BLA in our model, we implicitly include the effect of ACh in BLA by increasing the activity of the VIP cells, which then produce the low theta rhythm. Indeed, low theta in the BLA is known to depend on the muscarinic activation of CCK interneurons, a group of interneurons that overlaps with the class of VIP neurons in our model (Mascagni and McDonald, 2003; Krabbe et al., 2018).

Although the BLA can produce these rhythms, this does not rule out that other brain structures also produce the same rhythms through different mechanisms, and these can be transmitted to the BLA. Specifically, it is known that the olfactory bulb produces and transmits the respiratoryrelated low theta (4 Hz) oscillations to the dorsomedial prefrontal cortex, where it organizes neural activity (Bagur et al., 2021). Thus, the respiratory-related low theta may be captured by BLA LFP because of volume conduction or through BLA extensive communications with the prefrontal cortex. Furthermore, high theta oscillations are known to be produced by the hippocampus during various brain functions and behavioral states, including during spatial exploration (Vanderwolf, 1969) and memory formation/retrieval (Raghavachari et al., 2001), which are both involved in fear conditioning. Similarly to the low theta rhythm, the hippocampal high theta can manifest in the BLA. It remains to understand how these other rhythms may interact with the ones described in our paper. However, we emphasize that there is also evidence (as discussed above) that these rhythms arise within the BLA.”

**Reviewer #2 (Recommendations for the Authors):**
(1) Three different types of VIP interneurons with distinct firing patterns have been revealed in the BLA (Rhomberg et al., 2018). Does the generation of rhythmic activities depend on the firing features of VIP interneurons? Does it matter whether VIP interneurons fire burst of action potentials or they discharge more regularly?(2) The authors used data for modeling SST interneurons obtained e.g., in the hippocampus. However, there are studies in the BLA where the intrinsic characteristics of SST interneurons have been reported (Unal et al., 2020; Guthman et al., 2020; Vereczki et al., 2021). Have the authors considered using results of studies that were conducted in the BLA?

We thank the Reviewer for their questions, which have helped us further improve our manuscript in response to similar queries from Reviewer 3 in the previous review round. More in detail:

(1) Although other electrophysiological types exist (Sosulina et al., 2010), we hypothesized that the electrophysiological type of VIP neurons that display intrinsic stuttering is the type that would be involved in mediating low theta oscillations during fear conditioning. This is because VIP intrinsic stuttering in cortical neurons is thought to involve the D-current, which helps create low theta bursting oscillations in the neuronal spiking patterns (Chartove et al., 2020). We think that the other subtypes of VIP interneurons are not essential for the low theta oscillatory dynamics observed during fear conditioning and, thus, did not provide an essential constraint for the phenomena we are trying to capture. VIP interneurons in our network must fire bursts at low theta to be effective in creating the pauses in ECS and F spiking needed for potentiation; single spikes at theta are not sufficient to create these pauses.

(2) In our model, we used the results conducted in a BLA study (Sosulina et al., 2010). SOM cells in the BLA display several physiologic types. We chose to include in our model the type showing early adaptation in response to a depolarizing current and inward (outward) rectification upon the initiation (release) of a hyperpolarizing current. We hypothesize that this type can produce high theta oscillations, a prominently observed rhythm in the BLA. Unal et al., 2020 (Unal et al., 2020) found two populations of SOM cells in the BLA, which have been previously recorded in (Sosulina et al., 2010), including the one type we chose to model. This SOM cell type shows a low threshold spiking profile characterized by spike frequency adaptation and voltage sag indicative of an H-current used in our model. Guthman et al., 2020, (Guthman et al., 2020), also found a population of SOM cells with hyperpolarization induced sag.

Our model also uses a NaP-current for which there is no data in the BLA. However, it is known to exist in hippocampal SOM cells and that NaP- and H- currents can produce such a high theta in hippocampal cells. It is a standard practice in modeling to use the best possible replacement for unknown currents. Of course, it is unfortunate to have to do this. We also note that models can be considered proof of principle, that can be proved or disproved by further experimental work. Both (Guthman et al., 2020) and (Vereczki et al., 2021) also uncover further heterogeneity among BLA SOM interneurons involving more than electrophysiology. We hypothesize that such a level of heterogeneity revealed by these three studies is not key to the question we are asking (where crucial ingredients are the rhythms) and, therefore, was not included in our minimal model.

We modified the Discussion section titled “Assumptions and predictions of the model” as follows:

“Our model, which is a first effort towards a biophysically detailed description of the BLA rhythms and their functions, does not include the neuron morphology, many other cell types, conductances, and connections that are known to exist in the BLA; models such as ours are often called “minimal models” and constitute most biologically detailed models. For example, although there is considerable variability in the activity patterns of both VIP cells and SOM cells (Sosulina et al., 2010; Guthman et al., 2020; Ünal et al., 2020; Vereczki et al., 2021), our focus was specifically on those subtypes that generate critical rhythms within the BLA. Such minimal models are used to maximize the insight that can be gained by omitting details whose influence on the answers to the questions addressed in the model are believed not to be qualitatively important. We note that the absence of these omitted features constitutes hypotheses of the model: we hypothesize that the absence of these features does not materially affect the conclusions of the model about the questions we are investigating. Of course, such hypotheses can be refuted by further work showing the importance of some omitted features for these questions and may be critical for other questions. Our results hold when there is some degree of heterogeneity of cells of the same type, showing that homogeneity is not a necessary condition.”

(3) The authors may double-check the reference list, as e.g., Cuhna-Reis et al., 2020 is not listed.

We thank the Reviewer for spotting this. We checked the reference list and all the references are now listed.

Finally, we wanted to acknowledge that we made other changes to the manuscript unrelated to the reviewers’ questions with the purpose of gaining clarity. More specifically:

(1) We included a section titled “Significance” after the abstract and keywords, which reads as follows:

“Our paper accounts for the experimental evidence showing that amygdalar rhythms exist, suggests network origins for these rhythms, and points to their central role in the mechanisms of plasticity involved in associative learning. It is one of the few papers to address high-order cognition with biophysically detailed models, which are sometimes thought to be too detailed to be adequately constrained. Our paper provides a template for how to use information about brain rhythms to constrain biophysical models. It shows in detail, for the first time, how multiple interneurons help to provide time scales necessary for some kinds of spike-timing-dependent plasticity (STDP). It spells out the conditions under which such interactions between interneurons are needed for STDP and why. Finally, our work helps to provide a framework by which some of the discrepancies in the fear learning literature might be reevaluated. In particular, we discuss issues about Hebbian plasticity in fear learning; we show in the context of our model how neuromodulation might resolve some of those issues. The model addresses issues more general than that of fear learning since it is based on interactions of interneurons that are prominent in the cortex, as well as the amygdala.”

(2) The Result section “Physiology of the interneuron types is critical to their role in depression-dominated plasticity”, which is now titled “Mechanisms by which interneurons contribute to potentiation in depression-dominated plasticity”, now reads as follows:

“Mechanisms by which interneurons contribute to potentiation during depressiondominated plasticity. The PV cell is necessary to induce the correct pre-post timing between ECS and F needed for long-term potentiation of the ECS to F conductance. In our model, PV has reciprocal connections with F and provides lateral inhibition to ECS. Since the lateral inhibition is weaker than the feedback inhibition, PV tends to bias ECS to fire before F. This creates the fine timing needed for the depression-dominated rule to instantiate plasticity. If we used the classical Hebbian plasticity rule (Bi and Poo, 2001) with gamma frequency inputs, this fine timing would not be needed and ECS to F would potentiate over most of the gamma cycle, and thus we would expect random timing between ECS and F to lead to potentiation (Fig. S4). In this case, no interneurons are needed (See Discussion “Synaptic plasticity in our model” for the potential necessity of the depression-dominated rule).

In this network configuration, the pre-post timing for ECS and F is repeated robustly over time due to coordinated gamma oscillations (PING, as shown in Fig. 4A, Fig. 1C) arising through the reciprocal interactions between F and PV (Feng et al., 2019). PING can arise only when PV is in a sufficiently low excitation regime such that F can control PV activity (Börgers et al., 2005), as in Fig. 4A. However, although such a low excitation regime establishes the correct fine timing for potentiation, it is not sufficient to lead to potentiation (Fig. 4A, Fig. S2C): the depression-dominated rule leads to depression rather than potentiation unless the PING is periodically interrupted. During the pauses, made possible only in the full network by the presence of VIP and SOM, the history-dependent build-up of depression decays back to baseline, allowing potentiation to occur on the next ECS/F active phase. (The detailed mechanism of how this happens is in the Supplementary Information, including Fig. S2). Thus, a network without the other interneuron types cannot lead to potentiation. Though a low excitation level for a PV cell is necessary to produce a PING, a higher excitation level is necessary to produce a pause in the ECS and F. This higher excitation level is consistent with the experimental literature showing a strong activation of PV after the onset of CS (Wolff et al., 2014). The higher excitation happens when the VIP cell is silent, whereas a low excitation level is achieved when the VIP cell fires and partially inhibits the PV cell (Fig. 4B, Fig. S2D). The interruption in the ECS and F activity requires the participation of another interneuron, the SOM cell (Figs. 2B, S2): the pauses in inhibition from the VIP periodically interrupt ECS and F firing by releasing PV and SOM from inhibition and thus indirectly silencing ECS and F. Without these pauses, depression dominates (see SI section “ECS and F activity patterns determine overall potentiation or depression”).”

We also removed a supplementary figure (Fig. S2).

(3) We wanted to be clear and motivate our choice to extend the low theta range to 2-6 Hz and the high theta range to 6-14 Hz, compared to the 3-6 Hz and 6-12 Hz, respectively in the BLA experimental literature. Our main reason for extending the ranges was because the peaks of low and high theta power in the VIP and SOM cells, respectively, (the cells that generate these oscillations) occurred at the borders of the experimental ranges. Thus, in order to include the peaks of the model LFP, we lowered the low theta range by 1 Hz and increased the high theta range by 2 Hz.

We present a new supplementary figure (Fig. S1) containing the power spectra of VIP, which is the source of low theta in our model, and SOM interneuron, which is the source of high theta:

We mention Fig. S1 in the Result section “Rhythms in the BLA can be produced by interneurons”, where we added the following text: o “In the baseline condition, the condition without any external input from the fear conditioning paradigm (Fig. 1B, top), our VIP neurons exhibit short bursts of gamma activity (~38 Hz) at low theta frequencies (~2-6 Hz) (peaking at ~3.5 Hz) (see Fig. S1A).” o “In our baseline model, SOM cells have a natural frequency of ~12 Hz (Fig. 1B, middle; Fig. S1B), which is at the upper limit of the experimental high theta range; this motivates our choice to extend the high theta range up to 14 Hz in order to include the peak.”

Knowing the natural frequencies of VIP and SOM interneurons from the Result section “Rhythms in the BLA can be produced by interneurons”, we specified more clearly that we quantify the change of power in the low and high theta range around the power peaks in those ranges. Specifically, we changed some sentences in the first paragraph of the Result section “Increased low-theta frequency is a biomarker of fear learning” as follows:

“We find that fear conditioning leads to an increase in low theta frequency power of the network spiking activity compared to the pre-conditioned level (Fig. 6 A,B); there is no change in the high theta power. We also find that the LFP, modeled as the linear sum of all the AMPA, GABA, NaP-, D-, and H- currents in the network, similarly reveals a low theta power increase when considering the peak of the low theta power, and no significant variation in the high theta power again when considering the peak of the high theta power (Fig. 6 C,D,E).”

Finally, we made a few other small changes:

In the Introduction, we mention the following: “We also note that there is not uniformity on the exact frequencies associated with low and high theta, e.g., ((Lorétan et al., 2004) used 2-6 Hz for low theta). Here, we use 2-6 Hz for the theta range and 6-14 Hz for the high theta range.”

In Fig. 6DE (reported below point 3), we reran the statistics using a smaller interval for high theta (11.5-13 Hz) to focus around the peak. Our initial result showing significant change in low theta between pre and post fear conditioning and no change in high theta still holds.

In Fig. 6 of the Result section “Increase low-theta frequency is a biomarker of fear learning”, we switched the order of panels F and G. This change allows us to first focus on the AMPA currents, which are the major contributors of the low theta power increase, and to specify what AMPA current drives that increase. After that, we present the power spectrum of the GABA currents, as well.

The corresponding text in the Result section, now reads as follows:

“We find that fear conditioning leads to an increase in low theta frequency power of the network spiking activity compared to the pre-conditioned level (Fig. 6 A,B); there is no change in the high theta power. We also find that the LFP, modeled as the linear sum of all the AMPA, GABA, NaP-, D-, and H- currents in the network, similarly reveals a low theta power increase when considering the peak of the low theta power, and no significant variation in the high theta power again when considering the peak of the high theta power (Fig. 6 C,D,E). These results are consistent with the experimental findings in (Davis et al., 2017). Specifically, the newly potentiated AMPA synapse from ECS to F ensures F is active after fear conditioning, thus generating strong currents in the PV cells to which it has strong connections (Fig. 6F). It is the AMPA currents to the PV interneurons that are directly responsible for the low theta increase; it is the newly potentiated ECS to F synapse that paces the AMPA currents in the PV interneurons to go at low theta. Thus, the low theta increase is due to added excitation provided by the new learned pathway.”

(4) In the Discussion section “Assumptions and predictions of the model”, we specified the following:

“Our model predicts that blockade of D-current in VIP interneurons (or silencing VIP interneurons) will both diminish low theta and prevent fear learning. Finally, the model assumes the absence of significantly strong connections from the excitatory projection cells ECS to PV interneurons, unlike the ones from F to PV. Including those synapses would alter the PING rhythm created by the interactions between F and PV, which is crucial for fine timing between ECS and F needed for LTP.”

(5) Finally, to broaden the potential interest of our study, we added the following sentences:

At the conclusion of the abstract:

“The model makes use of interneurons commonly found in the cortex and, hence, may apply to a wide variety of associative learning situations.” - At the conclusion of the introduction:

“Finally, we note that the ideas in the model may apply very generally to associative learning in the cortex, which contains similar subcircuits of pyramidal cells and interneurons: PV, SOM and VIP cells.”

Also, changes in the emphasis of the paper led us to remove the following from the abstract: “Finally, we discuss how the peptide released by the VIP cell may alter the dynamics of plasticity to support the necessary fine timing.”